

# Drivers of long-term grassland CO₂ fluxes and regrowth: effects of management and meteorological conditions over time

Yi Wang, Iris Feigenwinter, Lukas Hörtnagl, Anna K. Gilgen, Nina Buchmann

Institute of Agricultural Sciences, ETH Zurich, 8092 Zurich, Switzerland

*Correspondence to*: Yi Wang (yi.wang@usys.ethz.ch)

**Abstract.** Grasslands serve a unique role in the global carbon (C) cycle and cover about 30% of the European and about 70% of the Swiss agricultural area. Carbon dioxide ($CO_2$) fluxes of managed grasslands are substantially influenced by land management practices and meteorological conditions, but the temporal development of these drivers is still uncertain. With the eddy covariance (EC) technique, net ecosystem $CO_2$ exchange (NEE) can be directly measured, and then partitioned into

gross primary production (GPP; amount of $CO_2$ fixed through photosynthesis) and ecosystem respiration (Reco; amount of $CO_2$ released via plant and soil respiration). We used 20 years (2005–2024) of EC fluxes, meteorological data, and detailed management information collected from an intensively managed grassland site (Chamau) in Switzerland, and employed machine learning approaches, i.e., eXtreme Gradient Boosting (XGBoost) models in combination with SHapley Additive exPlanations (SHAP) analyses, to identify drivers and their temporal contributions over two decades. Our study aimed to (1)

investigate intra- and inter-annual variations in grassland $CO_2$ fluxes, (2) assess magnitude and drivers of GPP and Reco during the regrowth periods (i.e., after mowing, grazing, or reseeding), and (3) quantify driver contributions to GPP and Reco over time, with focus on management and extreme events. $CO_2$ fluxes showed pronounced intra- and inter-annual variations, driven by both management activities as well as meteorological conditions. Despite significant increases in temperature and decreases in soil water content (SWC) during the two decades, GPP and Reco rates during regrowth periods

remained stable, and no significant trend over time was detected, suggesting adapted, climate-smart decision making of the farmer. The most important drivers of GPP in the long-term were light, management, and temperature, while Reco was mainly driven by temperature, GPP, and management. However, during extreme drought periods in the peak growing season (June, July, August), SWC increased in importance and limited GPP. In contrast, the impact of nitrogen (N) fertilization was more differentiated, either acting in parallel with SWC, suggesting low N availability during drought periods, or increasing

GPP in years after sward renewal despite low SWC. Overall, our study provided novel insights into relevant drivers of grassland $CO_2$ fluxes and their complex temporal contributions in the short- and long-term. Our results suggest that even small climate-smart management adaptations could be promising solutions for stabilizing important grassland processes, such as grassland regrowth, under current and future climate.

**Keywords.** Temperate grasslands; climate-smart farming; climate extremes; eddy covariance; driver contributions



## 1 Introduction

Grassland ecosystems cover 40% of the global ice-free land surface and contribute about one third to the terrestrial carbon (C) stocks (Bai and Cotrufo, 2022; White et al., 2000). Moreover, grasslands provide a broad set of ecosystem services and thus play a unique role in global and regional feed and food production (Henwood, 1998; Richter et al., 2024; Schils et al.,

2022; Wang et al., 2025b). Temperate permanent grasslands (i.e., grasslands continuously used for forage production, not being part of any crop rotation) account for about 34% of the European and about 70% of the Swiss agricultural areas (Eurostat, 2020; FSO, 2021). With great capacity to store C belowground and the large potential to act as a significant C sink, these agroecosystems are often considered as nature-based solutions to mitigate climate change (Bengtsson et al., 2019; Jia et al., 2019). Therefore, accurate understanding and assessment of grassland C dynamics and their drivers are of high

relevance to inform grassland management and policies (EASAC, 2022; European Commission, 2023; Soussana et al., 2004; Wang et al., 2019; Xia et al., 2015).

As an essential part of the grassland C cycle, carbon dioxide ($CO_2$) fluxes at the ecosystem scale are the result of two opposing fluxes: $CO_2$ uptake as gross primary production (GPP) via plant photosynthesis, and $CO_2$ release as ecosystem respiration (Reco) via autotrophic and heterotrophic respiration (Reichstein et al., 2005). The balance between these two

processes represents the net ecosystem $CO_2$ exchange (NEE) between the ecosystem and the atmosphere, which can be directly measured at high temporal resolution with the eddy covariance (EC) technique (Aubinet et al., 2012; Eugster and Merbold, 2015). Dynamics of grassland $CO_2$ fluxes are substantially driven by land management practices (e.g., mowing, grazing, fertilization, sward renewal), climate conditions (e.g., trends, variabilities, and anomalies), and their interactions (Ammann et al., 2007, 2020; Feigenwinter et al., 2023a; Liu et al., 2024; Rogger et al., 2022; Wall et al., 2019, 2020; Winck

et al., 2025, 2023; Zeeman et al., 2010). Yet, disentangling their respective effects remains challenging due to high variability across different grassland types, diverse management practices, as well as strong interdependencies and confounding interactions among drivers (Barneze et al., 2022; Chang et al., 2021; Shi et al., 2022).

With ongoing anthropogenic climate change, temperate regions are experiencing rising temperature and atmospheric dryness, reduced precipitation and soil moisture, along with more frequent, intense, and prolonged extreme events such as

droughts and heatwaves, and compound extremes (Fischer and Knutti, 2014; Grillakis, 2019; Hermann et al., 2024; IPCC, 2023; Shekhar et al., 2024; Zscheischler et al., 2014). Global meta-analyses showed that warming can promote grassland productivity and $CO_2$ fluxes in general (Shi et al., 2022; Wang et al., 2019), but temperate grasslands have shown higher sensitivity to warming for Reco compared to GPP (Wang et al., 2019). In contrast, a latitudinal, cross-site study showed GPP and Reco consistently increased with increasing precipitation but decreased with increasing temperature (Liu et al., 2024).

The effect of warming on $CO_2$ fluxes was also found to interact with management activities such as fertilization (Barneze et al., 2022), mowing or grazing (Shi et al., 2022; Zhou et al., 2019). While extreme events (here mainly droughts, heatwaves, and compound events) typically resulted in reduced magnitude of $CO_2$ fluxes (Fu et al., 2020; Li et al., 2023a; Reichstein et al., 2013; Zscheischler et al., 2014), their impacts can also be influenced by different management practices. A long-term



study comparing intensive and extensive grazing in France found that extensive management may buffer $CO_2$ fluxes against
temperature anomalies (Winck et al., 2025), while another observational study in Switzerland demonstrated that certain
management activities could result in higher than usual net $CO_2$ release, if followed by unfavourable meteorological
anomalies (Rogger et al., 2022). Given these diverse findings across spatial and temporal scales and under varying
management regimes, it remains difficult to pinpoint the dominant drivers of $CO_2$ fluxes or to determine which management
practices should be recommended to adapt to or to mitigate future climate risks in grassland-based farming systems.
Long-term field studies using the EC technique have been able to capture the temporal variability of $CO_2$ fluxes with
additional consideration of actual management activities (Ammann et al., 2020; Feigenwinter et al., 2023a; Rogger et al.,
2022; Winck et al., 2025). However, current studies evaluating the drivers of $CO_2$ fluxes have typically used traditional
linear models to analyze the effects of climate conditions, only sometimes considering also management practices. Linear
models frequently fail to capture the nonlinear, interactive, and highly dynamic nature of drivers that influence $CO_2$ fluxes in
complex systems such as intensively managed agroecosystems (Feigenwinter et al., 2023b; Maier et al., 2022). As a
powerful alternative, machine learning models are capable of integrating these complex drivers at high temporal resolution
and can account for management events. Although these models operate as black boxes, providing only aggregated measures
of driver importance, advancements in interpretable machine learning such as SHapley Additive exPlanations (SHAP)
analysis (Lundberg et al., 2020; Lundberg and Lee, 2017) now enable the estimation of individual driver contributions and
importance for each discrete observation. This capability allows the attribution of $CO_2$ flux variability and facilitates the
identification of temporal changes in underlying drivers. In this study, we present 20 years (2005–2024) of $CO_2$ fluxes
measured at an intensively managed temperate grassland in Switzerland. Using additional meteorological data and well-
documented management information, we applied eXtreme Gradient Boosting (XGBoost) models in combination with
SHAP analysis to address the following objectives: (1) investigate intra- and inter-annual variations in grassland $CO_2$ fluxes,
(2) assess magnitude and drivers of GPP and Reco during the regrowth periods (i.e., after mowing, grazing, or reseeding),
and (3) quantify driver contributions to GPP and Reco over time, with focus on management and extreme events.

## 2 Material and methods

### 2.1 Study site

The study site Chamau is a permanent grassland located in the canton of Zug, Switzerland (47°12'36.8″ N and 8°24'37.6″ E,
393 m a.s.l). In 2005, an eddy covariance and meteorological station (part of FLUXNET as CH-Cha) was established, and
$CO_2$ and $H_2O$ vapour fluxes have been continuously measured since then. The average annual temperature is 9.9 °C, and the
average annual precipitation sum is 1147 mm over the study period (2005–2024). The soil of the grassland parcels
surrounding the flux station is classified as a Cambisol/Gleysol and has a silt loam soil texture, with a pH of 5.3 in the
topsoil (Roth, 2006). This grassland is intensively managed, with four to six mowing events per year, occasional grazing by
sheep, and organic fertilizer application (mainly as slurry after mowing, on average 260 kg N ha$^{-1}$ yr$^{-1}$ from 2005 to 2024). In



February 2012 and August 2021, the sward was renewed, i.e., the existing sward was destroyed and ploughed, and a new sward was sown. The vegetation in the grassland includes English ryegrass (*Lolium perenne*), common meadowgrass (*Poa pratensis*), Italian ryegrass (*Lolium multiflorum*), white clover (*Trifolium repens*), red clover (*Trifolium pratense*), and dandelion (*Taraxacum officinale*) (Feigenwinter et al., 2023a).

**2.2 Management information and regrowth periods**

The Chamau grassland site consists of two parcels A and B (Feigenwinter et al., 2023a), and detailed management information from this site has been continuously documented since 2005, including major activities like mowing, grazing, fertilization, and sward renewal. An $N_2O$ mitigation experiment was carried out from 2015 to 2020, where parcel A received standard fertilization, while parcel B was oversown with a higher legume proportion and received no fertilization
(Feigenwinter et al., 2023b; Fuchs et al., 2018). Since parcel B dominated the footprint area in earlier years (Fig. B3 in Feigenwinter et al., 2023a), and the two parcels were under similar mowing and grazing scheme in recent years, we adopted the management scheme of parcel B to define regrowth periods in the current study.

We defined the period between two mowing/grazing/sward renewal events as a regrowth period (excluding the day(s) when management occurred). Each regrowth period consists of a minimum of ten days. This allowed us to consider slightly
different management activities on both parcels as one regrowth period. For example, in earlier years, the parcels were occasionally first partially grazed and then partially mowed within a certain time window, or vice versa. We thus considered such management patterns as one regrowth period. A 'DaySinceUse' variable was generated for each day in a regrowth period to indicate numbers of days since the last mowing/grazing/sward renewal event. An average daily fertilizer nitrogen (N) input for each regrowth period (DailyN, in kg N ha$^{-1}$ day$^{-1}$) was calculated based on the total fertilizer N received during
the period and the length of that period. If parcel A and B had different N inputs for certain regrowth periods, the average of the two parcels was used as the total N input. For each regrowth period, $CO_2$ fluxes, i.e., GPP and Reco regrowth rates (in g C m$^{-2}$ day$^{-1}$), were calculated based on the cumulative $CO_2$ uptake/release over the regrowth period (in g C m$^{-2}$) divided by the length of the regrowth period (i.e., number of days). If the middle date of one regrowth period was between April and September (which is the main growing season), the period was classified as a spring-summer period, else as an autumn-
winter period.

**2.3 Flux and meteorological measurements and data processing**

Since July 2005, molar densities of $CO_2$ and $H_2O$ vapour were measured with open path infrared gas analyzers (IRGAs; LI-7500; LI-COR Biosciences, USA) at 2.42 m measurement height, together with measurements of 3D wind speed and wind direction using ultrasonic anemometers (R3–50, Gill Instruments Ltd., UK) at 20 Hz time resolution. Meteorological
variables, including air temperature (TA, in °C), relative humidity (RH, in %), air pressure (PA, in hPa), incoming and outgoing shortwave and longwave radiation (SW IN/OUT and LW IN/OUT, in W m$^{-2}$), incoming and outgoing



photosynthetic photon flux density (PPFD IN/OUT, in $\mu mol\ m^{-2}\ s^{-1}$), precipitation (PREC, in mm), soil temperature (TS, in °C), and soil water content (SWC, %), were also measured. For detailed instrumentation, see Feigenwinter et al. (2023 a).

Half-hourly $CO_2$ and $H_2O$ vapour fluxes were calculated with EddyPro (v7.0.9) based on the covariance of the turbulent
fluctuations of gas concentration and vertical wind speed. Widely used community guidelines, including corrections for density fluctuations (Webb et al., 1980), double axis rotation (Wilczak et al., 2001), high-pass (Moncrieff et al., 2005) and low-pass filtering effects (Horst, 1997), were applied. Time lags were detected using covariance maximization in a small time window (between 0.05 s and 0.55 s), with a default time lag (typically 0.30 s) if no clear time lag was found. Post-processing for quality assessment and gap-filling was done in the Python library diive (Hörtnagl, 2025a), NEE flux
partitioning into GPP and Reco was done with the R package REddyProc (v1.3.3, Wutzler et al., 2018). During quality assessment, a series of quality control flags were applied to classify data quality (after testing steady-state, data completeness, spectral correction factor, IRGA signal strength, outlier detection), and turbulence-based constant friction velocity (USTAR) thresholds (0.05, 0.07, and 0.09 m $s^{-1}$ for the $16^{th}$, $50^{th}$, and $84^{th}$ percentile of the USTAR threshold distribution respectively) were applied. An overall Quality Control Flag (QCF) was generated integrating all assessments to
filter for high quality data. Resulting data gaps were then filled using random forest models utilizing a sliding three-year window for annual model training (Breiman, 2001; Hörtnagl, 2025a). For all our analysis, we used quality-controlled fluxes that were filtered with the $50^{th}$ percentile of the USTAR threshold distribution, subsequently gap-filled using random forest models, and partitioned into GPP and Reco based on the night-time method (Reichstein et al., 2005).

For meteorological data, a similar quality assessment procedure was applied with the diive library (Hörtnagl, 2025a), and all
screened variables were aggregated into half-hourly resolution. Vapour pressure deficit (VPD) was calculated based on TA and RH. For soil temperature (at 0.04 m, 0.15 m, and 0.4 m depths) and soil water content (at 0.05 m, 0.15 m, and 0.75 m depths) measurements, averages across all depths were calculated and used in the final analysis.

All data used in this study were derived from the most recent PI dataset (Hörtnagl et al., 2025), with all processing steps openly documented in a GitHub repository (Hörtnagl, 2025b).

**2.4 Data analysis**

**2.4.1 Trend analysis, extreme detection, and functional relationships**

To detect monotonic trends in monthly and yearly meteorological variables, we applied the Mann-Kendall test (Kendall, 1955; Mann, 1945). If the temporal trend was significant (p-value < 0.05), we additionally obtained the slope and sign of the trend with Sen's slope estimator (Sen, 1968). We tested trends in GPP and Reco across all regrowth periods, moreover, two
seasons were tested as well: spring-summer (April to September) and autumn-winter (October to March).

To identify months with high soil and air dryness, z-scores were calculated for monthly average SWC and VPD using the equation:

$$z = \frac{x - \mu}{\sigma} \qquad (1)$$



where $x$ represents the monthly average for SWC or VPD, $\mu$ is the respective overall mean and $\sigma$ is the respective standard

deviation for all monthly values across the entire study period. Months with z-scores for SWC lower than minus two were

defined as high soil dryness, and months with z-scores of VPD higher than two were defined as high air dryness.

To evaluate the functional relationship between PPFD and GPP, we assigned regrowth periods to three categories, i) two

years before (i.e., 2010 and 2011; 2019 and 2020) and ii) two years after (i.e., 2013 and 2014; 2022 and 2023) the grassland

renewal years (2012 and 2021), and iii) all remaining years without the renewal years. Second degree polynomial regressions

were then fitted using the average PPFD (calculated from the daily total PPFD) and GPP regrowth rates across all regrowth

periods within each category.

**2.4.2 XGBoost models and SHAP analysis**

To model effects of different drivers on daily GPP and Reco (in g C m$^{-2}$ day$^{-1}$), we applied two eXtreme Gradient Boosting

(XGBoost) models (Chen and Guestrin, 2016), one model for GPP and one model for Reco, based on 20 years of data. For

GPP, we selected management indicators (i.e., DaySinceUse and DailyN) and major meteorological variables (i.e., PPFD,

TA, TS, PREC, SWC, and VPD) as predicting features. For Reco, TS was used instead of TA, and GPP was added as an

additional feature. Input daily data from all 20 years were randomly split into 70% and 30% for training and testing datasets,

respectively. Major hyperparameters (i.e., number of estimators, max depth, learning rate, and subsample) in both XGBoost

models were fine-tuned using grid search with 10-fold cross-validation to improve performance and avoid overfitting. The

best hyperparameters that minimized root mean squared error were then used in the final models (Table A1). To evaluate

model performance and generalization on the testing dataset, metrics like root mean square error (RMSE) and the coefficient

of determination (R$^2$) were calculated.

To estimate the marginal contribution (i.e., SHAP value) of each feature to every model prediction (here, daily GPP and

Reco predictions), we used the TreeExplainer SHAP framework that was designed specifically for tree-based ensembles like

XGBoost (Lundberg et al., 2020). SHAP values at zero represent the overall mean prediction of the background dataset,

which is a collection of representative data points used to estimate the expected model output that SHAP compares

individual predictions against (Lundberg et al., 2020; Molnar, 2023). If a feature (i.e., a driver variable) has a positive SHAP

value, this feature increases the local prediction relative to the overall mean prediction, and vice versa. Higher absolute

SHAP values represent larger impacts (positive or negative) on predicted fluxes. For each prediction (here, daily GPP and

Reco), the sum of all SHAP values across all features equals the difference between the local model prediction (i.e., the

predicted fluxes) and the overall mean prediction (from the background dataset), which is consistent with the additive

property of SHAP values (Gou et al., 2024; Krebs et al., 2025; Qiu et al., 2022). In this study, we performed two SHAP

analyses (1 and 2) with different foci.

SHAP analysis 1: For a general comparison, the "tree_path_dependent" method was used as feature perturbation method.

Thus, the mean prediction at SHAP zero represents roughly the mean prediction (i.e., GPP or Reco) of the training dataset.





We then averaged daily SHAP values for each regrowth period. The mean of all SHAP values for one feature in a certain regrowth period was calculated to represent an average effect of that feature on predicted fluxes during this period.

SHAP analysis 2: To evaluate the impact of different drivers on daily GPP during the extreme months (detected in previous steps), we used the "interventional" method as feature perturbation method (Lundberg et al., 2020). Thus, SHAP values can

be calculated that compare the predictions to a certain subset of samples (Molnar, 2023). Here, we used only the months during the peak growing season (June, July, August) when no extreme weather conditions occurred as the background dataset and then calculated a second set of SHAP values for only the peak growing season. With this approach, we eliminated the impact of seasonality in the data and assessed only the impact of extreme events.

XGBoost models and SHAP analyses were performed in Python (version 3.11.11, Python Software Foundation, 2024) using

the libraries xgboost (v3.0.0, Chen and Guestrin, 2016), shap (v0.47.1, Lundberg and Lee, 2017), and scikit-learn (v1.6.1, Pedregosa et al., 2011). Other statistical analyses and visualization were done in R version 4.4.1 (R Core Team, 2024).

## 3. Results

### 3.1 Meteorological conditions and $CO_2$ fluxes over 20 years

During the past 20 years (2005–2024), the Chamau grassland experienced a mild temperate climate, with mean annual air

temperature of 9.9 C and mean annual precipitation of 1147 mm. At an annual scale, air temperature at this site increased significantly by 0.071 °C yr$^{-1}$ (Mann Kendall test p < 0.01), mean soil temperature and minimum soil temperature also increased by 0.070 °C yr$^{-1}$ and 0.095 °C yr$^{-1}$, respectively (p < 0.01). The years 2018 and 2022 were the hottest years on record, with mean annual air temperature of 11.0 °C and 11.1 °C, respectively, while the years 2010 and 2013 were the coldest years, with mean annual air temperature of 8.6 °C and 9.0 °C, respectively. Annual precipitation did not show a

significant trend (p = 0.87) over the 20 years. Nevertheless, the years 2018 and 2022 were the driest years on record, with annual precipitation of 906 mm and 884 mm, respectively, while the years 2016 and 2024 were the wettest years, with annual precipitation of 1351 mm and 1378 mm, respectively.

At the monthly scale, meteorological conditions showed pronounced seasonal variability (Fig. 1a-d). Overall, monthly air temperature reached maximum values in July (mean of 19.6 °C) and minimum values in January (mean of 1.0 °C). Monthly

air temperature (Fig. 1a), precipitation (Fig. 1b), and VPD (Fig. 1d) did not show significant trends over time (p = 0.21 for TA, p = 0.69 for PREC, p = 0.51 for VPD), while soil water content (Fig. 1d) decreased significantly over the 20 years (slope = -0.016 % month$^{-1}$, p < 0.01). Looking at each specific month (i.e., across 20 years), air and soil temperature increased significantly in the winter months (December to March; Table 1). August was the only month to experience a decrease in precipitation, which otherwise remained stable. Soil water content significantly decreased in late summer and

autumn (August to October; Table 1). Based on the z-scores, four months of extreme air dryness, seven months of extreme soil dryness, and six months of compound extremes (i.e., high VPD and low SWC) were identified (Fig. 2). These extreme





conditions mainly occurred between May and September, particularly during the peak growing season months (June, July, and August) in the years 2018, 2019, 2022 and 2023.

Figure 1. Meteorological variables and CO₂ fluxes at the Chamau grassland site from 2005 to 2024: (a) monthly mean air temperature (TA, red solid line) and soil temperature (TS, orange dashed line), (b) monthly sum of precipitation (PREC), (c) monthly mean daily total photosynthetic photon flux density (PPFD$_{sum}$), (d) monthly mean soil water content (SWC, blue dashed line) and vapour pressure deficit (VPD, purple solid line), (e) daily sum of net ecosystem CO₂ exchange (NEE, blue line), gross primary production (GPP, green line), and ecosystem respiration (Reco, orange line).

Daily NEE, GPP, and Reco showed strong interannual and seasonal variations over the 20 years (Fig. 1e, Table A2). Annual cumulative NEE ranged from 140 g C m⁻² yr⁻¹ (in 2012) to -503 g C m⁻² yr⁻¹ (in 2014), with an overall mean (± standard deviation) of -260 (± 172) g C m⁻² yr⁻¹ (Fig. A1). All three flux components were strongly influenced by management practices (Fig. A2, Table A3, A4) such as mowing and grazing. Strongest net CO₂ uptake occurred in early spring (March-April), while the largest net CO₂ release happened in early winter (December-January).



**Table 1. Monthly meteorological variables from 2005 to 2024. For each month, mean and standard deviation, trends (unit yr$^{-1}$; in red positive trends, in blue negative trends) and significance from Mann-Kendall test (p-values < 0.05: *, < 0.01: **) are shown.**

| Variable | Unit | Jan Mean±SD | Jan Trend | Feb Mean±SD | Feb Trend | Mar Mean±SD | Mar Trend | Apr Mean±SD | Apr Trend |
|---|---|---|---|---|---|---|---|---|---|
| TA_mean | °C | 0.96±1.95 | | 1.84±2.41 | 0.2* | 5.26±1.3 | | 9.48±1.36 | |
| TA_min | °C | -10.22±3.58 | | -8.96±3.74 | | -6.48±3.03 | 0.253* | -3.05±1.56 | |
| TA_max | °C | 12.45±3.36 | | 15.01±3.48 | | 19.8±2.26 | | 24.35±2.33 | |
| TS_mean | °C | 3.9±1.23 | 0.144** | 3.79±1.48 | 0.164** | 6.16±1.18 | 0.121* | 10.07±0.73 | |
| TS_min | °C | 2.24±1.05 | 0.096** | 1.97±1.14 | 0.11** | 3.43±1.47 | 0.141* | 6.49±1.27 | |
| TS_max | °C | 6.15±1.56 | 0.162* | 6.33±1.57 | 0.162* | 9.81±1.34 | | 13.84±1.54 | |
| PREC_sum | mm | 64.28±35.59 | | 48.81±23.57 | | 65.68±34.35 | | 83.51±48.92 | |
| SWC_mean | % | 48.19±2.19 | | 48.07±2.16 | | 47.65±2.99 | -0.236* | 45.15±3.85 | |
| SWC_min | % | 46.21±2.49 | | 46.3±2.94 | | 45.39±3.51 | -0.302* | 41.08±4.96 | |
| SWC_max | % | 49.76±2.19 | | 49.57±2.25 | | 49.07±2.47 | | 46.77±4.01 | |
| VPD_mean | hPa | 0.73±0.36 | | 1.2±0.39 | | 2.36±0.61 | | 3.8±1.17 | |
| VPD_max | hPa | 6.55±2.02 | | 10.13±3.51 | | 15.6±3.16 | | 21.28±4.9 | |

| Variable | May Mean±SD | May Trend | Jun Mean±SD | Jun Trend | July Mean±SD | July Trend | Aug Mean±SD | Aug Trend |
|---|---|---|---|---|---|---|---|---|
| TA_mean | 13.77±1.51 | | 18.1±1.11 | | 19.55±1.35 | | 18.61±1.45 | 0.157** |
| TA_min | 1.47±2.08 | | 5.9±1.64 | | 8.41±1.37 | | 7.56±1.09 | |
| TA_max | 28.18±2.56 | | 32.28±1.93 | | 33.71±1.7 | | 32.42±2.63 | |
| TS_mean | 13.83±1.06 | | 17.91±1.07 | | 20.08±1.15 | | 19.62±1.04 | 0.092* |
| TS_min | 10.58±1.14 | | 14.02±1.39 | | 17.1±0.8 | | 16.85±1 | |
| TS_max | 17.73±1.83 | | 22.1±1.99 | | 23.5±1.92 | | 22.91±1.67 | |
| PREC_sum | 135.84±49.35 | | 140.56±52.47 | | 148.2±71.1 | | 142.83±46.43 | -4.057* |
| SWC_mean | 44.46±3.89 | | 42.78±5.34 | | 42.17±5.83 | | 41.74±5.69 | -0.468* |
| SWC_min | 39.28±4.92 | | 38.13±5.85 | -0.356* | 37.16±5.45 | | 37.25±5.69 | |
| SWC_max | 45.94±3.97 | | 44.66±5.25 | | 44.34±5.37 | | 44.29±5.02 | |
| VPD_mean | 4.5±1.26 | | 6.21±1.41 | | 6.9±1.69 | | 5.43±1.4 | |
| VPD_max | 25.57±4.74 | | 32.02±4.52 | | 35.82±6.37 | | 29.97±7.05 | |





**Table 1. (continued)**

| Variable | Sep | | Oct | | Nov | | Dec | |
|---|---|---|---|---|---|---|---|---|
| | Mean±SD | Trend | Mean±SD | Trend | Mean±SD | Trend | Mean±SD | Trend |
| TA_mean | 14.91±1.4 | | 10.36±1.32 | | 5.01±1.06 | | 1.54±1.44 | |
| TA_min | 3.72±1.65 | | 0.26±2.49 | | -4.75±3.01 | 0.252* | -8.73±3.91 | |
| TA_max | 28.4±1.76 | | 23.56±2.28 | | 16.96±2.84 | | 12.41±1.98 | |
| TS_mean | 16.96±1.03 | | 13.41±0.81 | | 9.02±1.12 | 0.102* | 5.41±1.24 | 0.123** |
| TS_min | 13.91±1.17 | | 10.46±1.57 | 0.128* | 6±1.79 | 0.18* | 3.69±1.2 | 0.174** |
| TS_max | 20.35±1.79 | | 16.46±1.05 | | 12.1±1.2 | | 7.54±1.36 | 0.119* |
| PREC_sum | 93.18±44.69 | | 76.67±28.61 | | 66.45±44.14 | | 81.25±44.07 | |
| SWC_mean | 42.43±5.16 | -0.44* | 43.79±4.26 | -0.399* | 45.74±3.34 | | 47.72±3.05 | |
| SWC_min | 38.64±5.09 | -0.538* | 40.98±4.64 | | 43.62±3.41 | | 45.86±3.07 | |
| SWC_max | 44.85±4.57 | -0.351* | 46.01±3.8 | -0.306* | 47.33±3.31 | | 49.24±3.02 | |
| VPD_mean | 3.28±0.65 | | 1.54±0.52 | | 0.76±0.37 | -0.032* | 0.53±0.34 | |
| VPD_max | 21.64±3.8 | | 14.92±4.08 | | 8.48±2.87 | | 5.51±1.27 | |

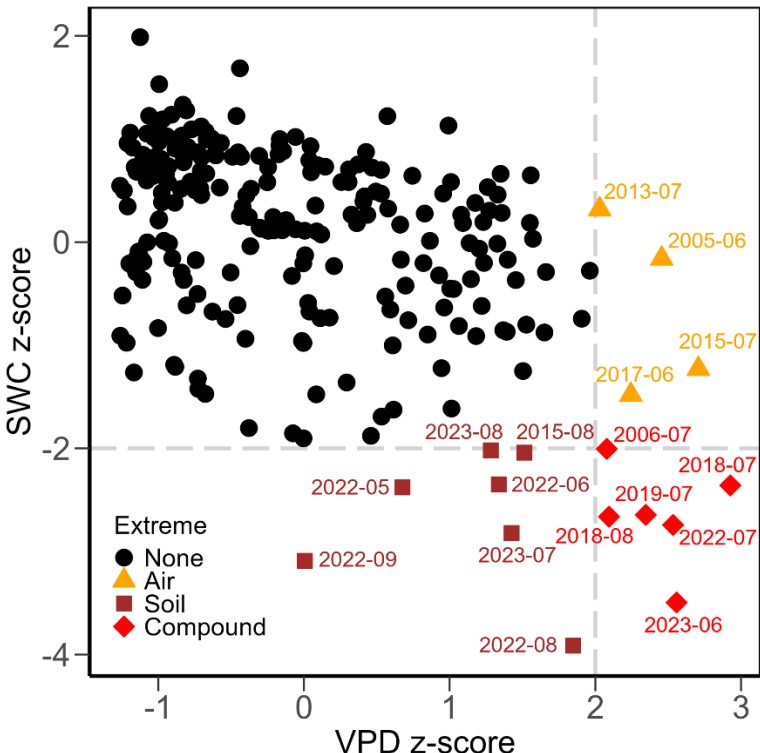

**Figure 2. Z-scores for monthly mean vapour pressure deficit (VPD) and soil water content (SWC) from 2005 to 2024. Grey dashed lines indicate thresholds for extreme months (z-score of 2 for VPD and of -2 for SWC). Symbols and colours indicate types of**
**extreme: none (black dots), high air dryness (orange triangles), high soil dryness (dark red squares), and compound air and soil dryness (red diamonds).**



## 3.2 Management and regrowth rates

The Chamau grassland was managed intensively during the past two decades, and experienced in total 94 mowing events, 319 days of grazing, and 102 fertilizer applications. Moreover, the grassland was renewed twice, in 2012 and 2021 (Fig. A2). This resulted in 115 grassland regrowth periods during the 20 years of this study. Overall, mean (± standard deviation) GPP regrowth rate was 7.2 (± 2.6) g C m$^{-2}$ day$^{-1}$ and Reco regrowth rate was 6.6 (± 2.5) g C m$^{-2}$ day$^{-1}$ (Fig. 3). During spring and summer seasons (i.e., April to September), about four mowing/grazing events took place (about two to three events in extreme dry years of 2018 and 2019), with a typical regrowth period of about 39 days. The average GPP regrowth rate during spring-summer seasons was 8.6 (± 1.4) g C m$^{-2}$ day$^{-1}$, while Reco regrowth rate averaged 7.9 (± 1.6) g C m$^{-2}$ day$^{-1}$. During autumn-winter seasons, regrowth rates were lower, with 4.1 (± 1.4) g C m$^{-2}$ day$^{-1}$ for GPP and 3.7 (± 1.5) g C m$^{-2}$ day$^{-1}$ for Reco. We did not observe any significant trend for GPP regrowth rates over the 20 years (overall p = 0.13), regardless of seasonality (spring-summer season: p = 0.10, autumn-winter season: p = 0.33). Similarly, Reco regrowth rates did not change significantly over time (overall p = 0.07), nor during the spring-summer season (p = 0.08) or the autumn-winter season (p = 0.84).

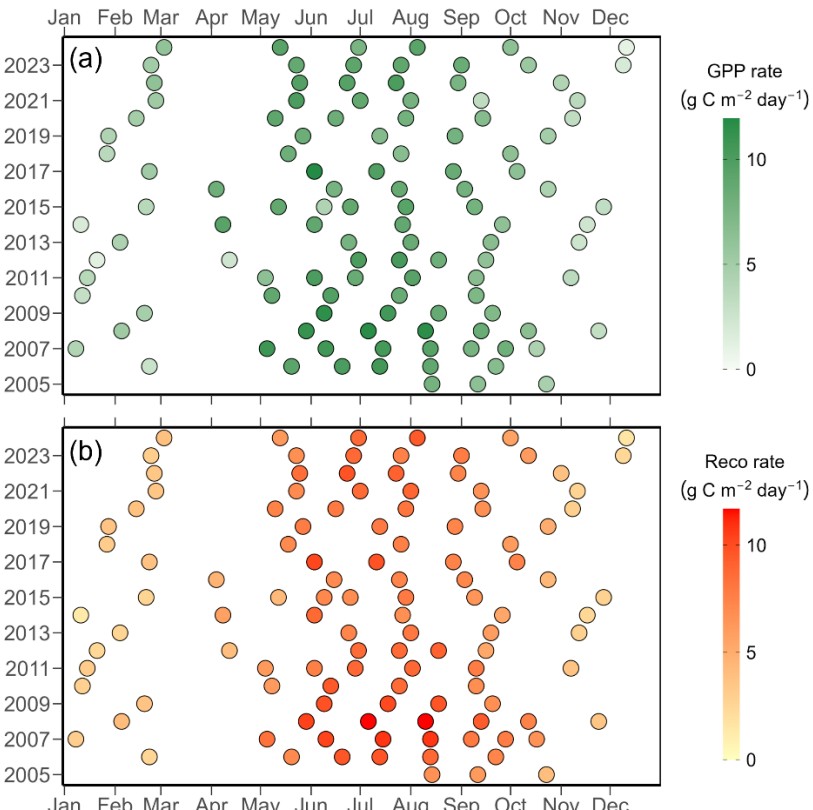

**Figure 3. Flux rates for (a) gross primary production (GPP) and (b) ecosystem respiration (Reco) for individual regrowth periods between 2005 and 2024. Points indicate the regrowth rates over a given regrowth period, with the x-axis position corresponding to the midpoint date of that period. Darker colours represent high regrowth rates.**



### 3.3 Drivers of GPP and Reco

The XGBoost models for daily GPP and Reco performed well (Fig. A3), with a RMSE for GPP of 1.2 g C m$^{-2}$ day$^{-1}$ and an R$^2$ of 0.91 on the testing set (Fig. A3a). The performance of the Reco models was better, with a RMSE of 0.6 g C m$^{-2}$ day$^{-1}$ and R$^2$ of 0.96 on the testing set (Fig. A3c). The three most important drivers for daily GPP were PPFD, TS, and DaySinceUse (Fig. A3b), while the three most important drivers for daily Reco were TS, GPP, and DailyN (Fig. A3d).

For both GPP and Reco, the drivers showed strong seasonality over the 20 years of the study (Fig. 4; SHAP analysis 1).

PPFD and temperature normally increased GPP relative to the mean in summer, while low light and low temperature conditions in winter decreased GPP from the mean (Fig. 4a). Compared to the main drivers of GPP (light, temperature, and management), water-related drivers (i.e., SWC, VPD, and PREC) had a relatively smaller contribution and showed a less strong seasonality. However, during the extreme summers (e.g., 2018, 2019, 2022, and 2023), especially SWC increasingly limited GPP (negative SHAP values compared to normal years). The main drivers of Reco, i.e., temperature and GPP,

showed a similar seasonality, with higher soil temperature and higher GPP in summer generally promoting respiration, while in winter, these drivers limited Reco (Fig. 4b). Compared to the drivers of GPP, fertilizer N inputs were more important for Reco than DaySinceUse, while water-related drivers had relatively low importance and low impacts on Reco. Negative effects of SWC on Reco during the extreme summers were more obvious in 2022 and 2023 compared to earlier years (e.g., 2018 and 2019).



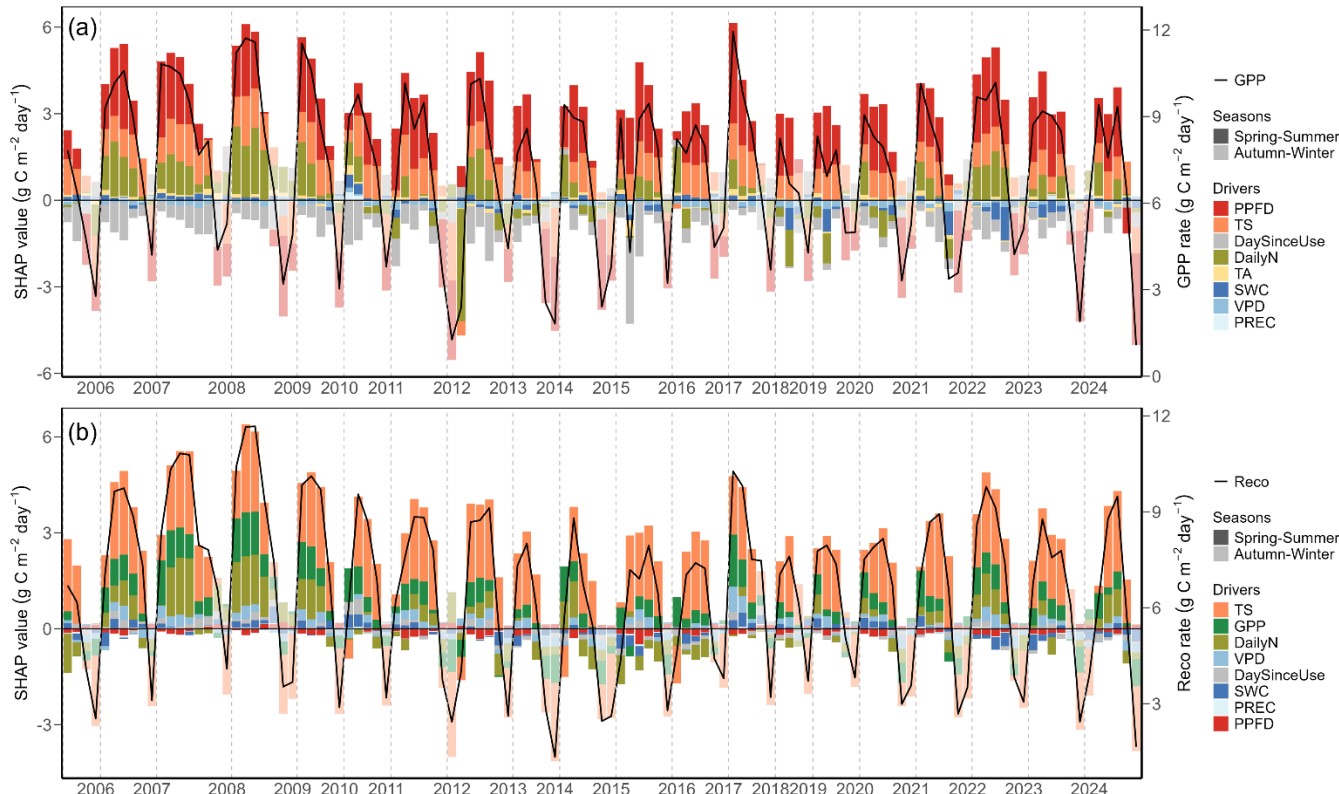

**Figure 4. Drivers of (a) gross primary production (GPP) and (b) ecosystem respiration (Reco) for all regrowth periods during the years 2005 to 2024 from SHAP analysis 1. Each stacked bar represents mean SHAP values of drivers from the overall XGBoost model, presented for each regrowth period: total photosynthetic photon flux density (PPFD, red), soil temperature (TS, orange), air temperature (TA, yellow), soil water content (SWC, dark blue), vapour pressure deficit (VPD, blue), precipitation (PREC, light blue), daily fertilizer nitrogen input (DailyN, moss green), day since last mowing, grazing or renewal (DaySinceUse, grey), and gross primary production (GPP, green). The mean prediction at SHAP value zero corresponds to a GPP of 6.1 g C m$^{-2}$ day$^{-1}$ and Reco of 5.3 g C m$^{-2}$ day$^{-1}$. Positive SHAP values represent positive impacts of drivers on GPP and Reco relative to the mean prediction, while negative values indicate negative impacts from the mean prediction. The regrowth rates of each regrowth period are shown as black lines. Bars with full opaque colours indicate regrowth periods in the spring-summer season (April-September). Bars with more transparent colours indicate regrowth periods in the autumn-winter season (October-March).**

## 3.4 Functional relationships between PPFD and GPP

Focusing on the years with exceptional management during the 20 years of this study, i.e., the renewal years 2012 and 2021, we tested the functional relationships between PPFD and GPP for the two years before and after the renewal years vs. all other normal years (Fig. 5). During all periods, daily GPP increased with mean daily PPFD before saturating at about 30 mol m$^{-2}$ day$^{-1}$ (before the renewal years) or at about 45 mol m$^{-2}$ day$^{-1}$ (after the renewal years, all other normal years). Moreover, spring-summer seasons in all years showed rather similar average GPP regrowth rates: 8.2 g C m$^{-2}$ day$^{-1}$ before the renewal years, 8.6 g C m$^{-2}$ day$^{-1}$ after the renewal years, and 8.9 g C m$^{-2}$ day$^{-1}$ for normal years. However, maximum GPP (GPPmax, i.e. the maximum of the curves in Fig. 5) was lower before the renewal years (8.5 g C m$^{-2}$ day$^{-1}$) compared to normal years



(9.7 g C m$^{-2}$ day$^{-1}$) and years after renewal (9.3 g C m$^{-2}$ day$^{-1}$), despite the fact that both years after the second renewal (2022

and 2023) experienced more frequent extreme weather conditions during summer (Fig. 2).

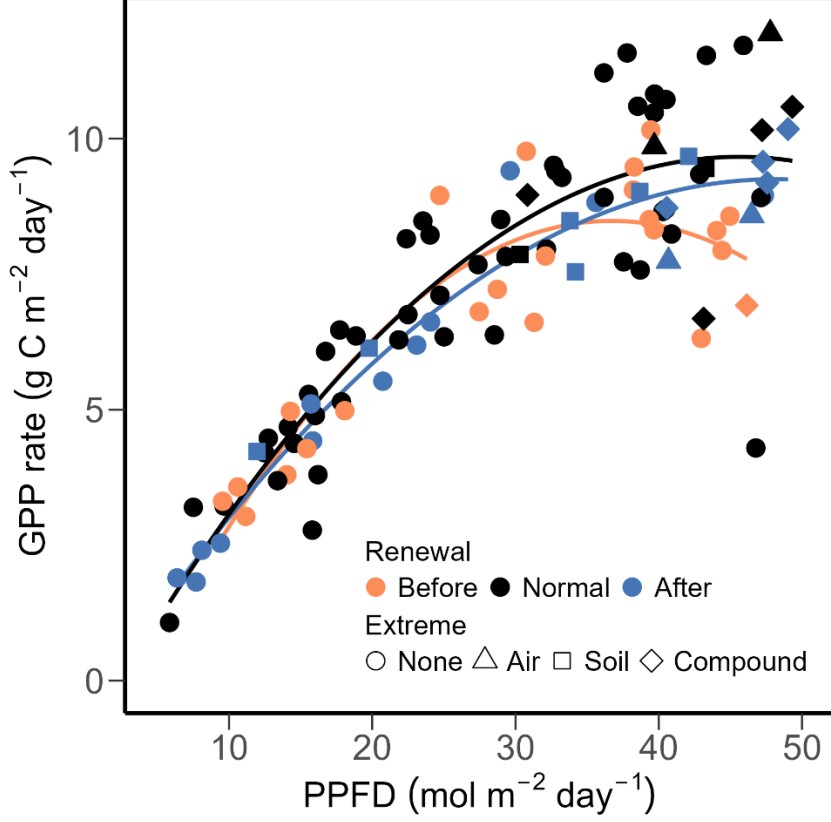

**Figure 5. Relationships between mean photosynthetic photon flux density (PPFD) and regrowth rate of gross primary production (GPP) for regrowth periods between 2005 and 2024, excluding the renewal years (2012 and 2021). Two years before the renewal years (i.e., 2010 and 2011; 2019 and 2020) are presented in orange, two years after the renewal years (2013 and 2014; 2022 and**

**2023) are shown in blue, while the remaining years are shown in black. Regrowth periods that spanned across extreme months (as shown in Figure 2) are shown in different shapes: high air dryness (triangles), high soil dryness (squares), and compound air and soil dryness (diamonds). Second-degree polynomial fits are shown in solid lines, with the same colour code as for the years.**

### 3.5 Temporal development of GPP drivers during extreme months

Focusing on the extreme summer months during the peak growing season, i.e., June, July, and August in the years 2018,

2019, 2022 and 2023 (lower right hand quadrant in Fig. 2; SHAP analysis 2), we found normal daily PPFD (Fig. 6a), but

significantly higher-than-normal daily temperature and VPD, accompanied by lower-than-normal soil water content (Fig. 6b-

c). Daily GPP during the peak growing season in 2018 and 2019 was smaller than the mean predicted GPP of non-extreme

months during the peak growing season (9.7 g C m$^{-2}$ day$^{-1}$), while GPP in 2022 and 2023 was more dynamic, ranging from

1.3 to 15.5 g C m$^{-2}$ day$^{-1}$ (Fig. 6d). Mowing and fertilization events were also more frequent in the recent years (2022, 2023)

compared to earlier years (2018, 2019), accompanied by higher amounts of N input. In contrast to the findings from all 20



years (Fig. 4), management-related drivers, i.e., DailyN and DaySinceUse, gained higher importance during these extreme summer months, while PPFD lost importance (Fig. 6d). In addition, soil water content surpassed soil temperature in importance, and its limiting impact on GPP was clearly seen in all four extreme summers, stronger in 2018 and 2019 than in 2022 and 2023. However, impacts of fertilizer N inputs on GPP differed during the four summers: In 2018 and 2019 (years

before the renewal in 2021), N fertilization impacts acted similarly to SWC (both drivers having negative SHAP values), suggesting low N availability during low SWC periods, while in 2022 and 2023 (years after the renewal in 2021), fertilizer N inputs increased GPP (positive SHAP values) despite low SWC values.

The relationship between SWC and its SHAP values for GPP during those four extreme summers showed a clear trend toward limitation of GPP at low SWC (negative SHAP values; Fig. A4). This limitation of GPP by SWC was stronger in the

years 2018 and 2019 (lowest SHAP values) compared to the years 2022 and 2023.



**Figure 6. Meteorological conditions and drivers of gross primary production (GPP) for days in the peak growing season (June, July, and August) during the years 2018, 2019, 2022 and 2023 from SHAP analysis 2. (a) Daily total photosynthetic photon flux density (PPFD$_{sum}$), (b) daily mean air temperature (TA, yellow line) and soil temperature (TS, orange line), (c) daily mean soil water content (SWC, dark blue line) and vapour pressure deficit (VPD, light blue line), (d) daily GPP (black dot-line) and SHAP values of each driver (stacked bar). Colours of the bars represent different drivers: photosynthetic photon flux density (PPFD, red), soil temperature (TS, orange), air temperature (TA, yellow), soil water content (SWC, dark blue), vapour pressure deficit (VPD, blue), precipitation (PREC, light blue), day since last mowing, grazing or renewal (DaySinceUse, grey), and daily fertilizer nitrogen input (DailyN, moss green). The mean prediction at SHAP value zero corresponds to a GPP of 9.7 g C m$^{-2}$ day$^{-1}$. Positive SHAP values represent positive impacts of drivers on GPP relative to the mean prediction, while negative values indicate negative impacts from the mean prediction. Symbols in the top of panel (d) represent management events during these months: mowing (squares) and fertilization (triangles). Numbers under the triangles give the amount of total fertilization nitrogen (N) input. The x-axis position of each symbol indicates the date of each event.**





## 4. Discussion

### 4.1 Intra- and interannual variations of $CO_2$ fluxes and management during 20 years


Overall, $CO_2$ fluxes (NEE, GPP, and Reco) at the Chamau grassland site showed significant intra- and inter-annual variation (Fig. 1e, Fig. A1), with sometimes higher inter-annual variation ($-260 \pm 172$, $2193 \pm 263$, $1934 \pm 244$ g C m$^{-2}$ yr$^{-1}$ for NEE, GPP and Reco respectively) than reported earlier for managed grasslands (e.g., Ammann et al., 2020; Heimsch et al., 2021, 2024; Peichl et al., 2011; Rogger et al., 2022). Reasons for these differences are manifold, considering that our study

spanned two decades of measurements (2005 to 2024). Besides short-term changes introduced from meteorological conditions and frequent management events, we also captured the impacts of multiple extreme periods, particularly dry and hot summer months in 2018, 2019, 2022, and 2023 (Fig. 2), a six-year $N_2O$ experiment with reduced organic fertilization (2015 to 2020; Feigenwinter et al., 2023b), as well as infrequent management practices such as two complete sward renewals (in 2012 and 2021; Feigenwinter et al., In prep; Merbold et al., 2014), all contributing to the highly dynamic nature of

ecosystem $CO_2$ fluxes observed at our site.

With climate change, GPP of terrestrial vegetation has been observed and predicted to increase (Keenan et al., 2014; Knauer et al., 2023; Myneni et al., 1997; Piao et al., 2007) due to $CO_2$ fertilization, increasing temperature, and longer growing seasons favouring photosynthesis, unless other resources become limiting, such as water or nutrients (He et al., 2017; Keenan et al., 2023; Madani et al., 2020; Terrer et al., 2019; Xu et al., 2019). Often, studies focused on natural ecosystems or

forests at regional and global scales (Anav et al., 2015; Cai and Prentice, 2020; Davi et al., 2006; Norby et al., 2010), but studies on grasslands also observed increasing trends in GPP and Reco with higher temperature in the past (Shi et al., 2022; Wang et al., 2019). These studies were often based on experiments in less intensively managed or even unmanaged grasslands, consequently with less intervention potential for farmers or land managers. In contrast, despite significant increases in mean annual temperature of around 1.4 °C over the past 20 years and decreases in soil water content during late

summer and autumn months (Table 1), we did not observe significant trends in GPP nor Reco of our intensively managed grassland (Fig. 3). This clearly indicates that despite observed climate change, the farmer managed to maintain regrowth rates and forage production over the past two decades, also seen in the harvest exports reported by Feigenwinter et al. (2023a) for the same site ($335 \pm 42$ g C m$^{-2}$ yr$^{-1}$ between 2005 and 2020). The local farmer adapted management practices, e.g., skipped individual harvests in extreme years (e.g., 2018), thereby extending the regrowth periods (Fig. A2). These

detailed adjustments are typically difficult to capture by standard large-scale models and coarse-resolution management datasets, demonstrating that management information at high spatio-temporal resolution is crucial to reliably model GPP of agricultural systems. However, management data at high temporal resolution is difficult to obtain, therefore large-scale modelling studies often use harmonized datasets at annual time scales (Winkler et al., 2021), potentially introducing substantial bias in modelled GPP.

In addition, infrequent management events such as sward renewals, typically carried out every 5-15 years depending on vegetation composition and soil texture (Schils et al., 2007; SUPER-G, 2018), are typically not considered in C cycle





models, partially due to lack of information about their occurrence but also simply due to lack of data for such events. For our site, two renewal events in 2012 and 2021 were included in the long time series (Fig. 1e, A1), with very different effects on NEE: while the event in 2012 led to a strong net $CO_2$ loss (cumulative NEE of 139 g C m$^{-2}$ yr$^{-1}$; Fig. A1), the event in
2021 resulted only in a weaker-than-normal net $CO_2$ uptake (cumulative NEE of -163 g C m$^{-2}$ yr$^{-1}$; Fig. A1). The two renewal events differed in seasons (February 2012 versus August 2021) and soil disturbance intensities (ploughed at 20 cm in 2012 and 3-4 cm in 2021), which subsequently influenced the establishment and regrowth of the new sward, likely explaining the observed difference in NEE. Other studies in managed grasslands have also reported different magnitudes of $CO_2$ fluxes after renewal, influenced by timing, intensity, and environmental conditions of the renewal events (Ammann et
al., 2020; Drewer et al., 2017; Li et al., 2023b). Such observations, albeit rare, but impactful in terms of C dynamics, underscore the necessity to include such infrequent destructive management practices in long-term flux studies as well as modelling frameworks.

## 4.2 Temporal development of driver contributions during regrowth periods using machine learning

NEE represents a combination of two large opposing fluxes, i.e., GPP and Reco, based on two physiological processes, i.e.,
photosynthesis and respiration, which respond differently to changing environmental conditions (Feigenwinter et al., 2023a; Peichl et al., 2011; Schwalm et al., 2010; Wang et al., 2019). Thus, using two separate machine learning models for GPP and Reco (XGBoost models with high performance, $R^2 > 0.9$ on testing datasets), we identified the main GPP and Reco drivers as well as the temporal development of daily driver contributions (SHAP analysis 1). In contrast to other studies, we not only included high resolution meteorological data but also information about management events in our analyses at daily time
scales. Consistent with previous studies (Gilmanov et al., 2007; Wohlfahrt et al., 2008), PPFD and temperature were the main drivers of GPP and Reco, respectively (Fig. 4, A3). However, management-related variables such as the number of days since the last management event and fertilizer N inputs were always among the top four to five important drivers for daily GPP and Reco, respectively, highlighting their importance for a mechanistic understanding of C dynamics and greenhouse gas fluxes in managed grasslands (Hörtnagl et al., 2018). The machine learning analyses also picked up the
smaller effects of fertilizer N inputs as a driver for GPP during the $N_2O$ mitigation experiment (2015 to 2020), when organic fertilization – and thus N inputs – was replaced with higher fractions of legumes for half of the area (Feigenwinter et al., 2023b; Fuchs et al., 2018). Since the majority of temperate grasslands in Europe are actively managed (Eurostat, 2023), long-term and detailed management information, combined with more field observational measurements such as EC fluxes, can support improved management strategies to enhance grassland ecosystem services under changing climate (Wang et al.,
2025b).

Focusing on GPP and its most important driver PPFD, we further examined the physiological consequences of the two renewals in 2012 and 2021. Although the sward had shown a lower performance before the renewals, i.e., lower GPPmax (orange line, Fig. 5), compared to the following years with higher GPPmax (black and blue lines), the slopes of the light-response curves were very similar. Differences in GPPmax resulted in high net $CO_2$ uptake rates after both renewals (average





cumulative NEE of -465 g C m$^{-2}$ yr$^{-1}$ for 2013, 2014, 2022, and 2023) and also higher yields than before the renewals (Table A3, A4). Thus, the farmer's decision to renew the grassland with a multi-species mixture was substantiated. Furthermore, frequent oversowing (Table B1 and B2 in Feigenwinter et al., 2023a) might further help to maintain the preferred species composition and a stable yield. However, even though oversowing more diverse mixtures (Schaub et al., 2020) or increasing the legume proportions (Fuchs et al., 2018) can also improve the yield quality and are less destructive management

interventions than sward renewals, they can be costly (Schaub et al., 2021) and positive results are not always guaranteed. Therefore, adopting a combination of these observed management practices is a crucial first step towards more climate-smart farming practice, helping to ensure stable production and sustain ecosystem services over the long term.

Besides consistent positive effects of light and temperature on GPP (Fig. 4; red and orange bars), fertilizer N inputs (moss green bars) also promoted GPP during spring-summer seasons. Moreover, our analysis demonstrated the complex

interactions between N inputs and soil water availability: the positive impacts of N on GPP diminished in drier years (e.g., 2018, 2019) or even became negative, indicating lower soil N availability and thus lower N uptake for GPP when soil moisture was low (Bloor and Bardgett, 2012; Hartmann et al., 2013). For Reco, negative effects of N inputs were more obvious during the spring-summer seasons 2015 to 2020, i.e., when the N$_2$O mitigation experiment was ongoing, and the entire site received around half of fertilizer N input compared to other years (Feigenwinter et al., 2023b; Fuchs et al., 2018).

Without N inputs via organic fertilizer (mostly slurry), the organic C inputs and the C sink strength were reduced as well (Feigenwinter et al., 2023a). This resulted in less C available for microbial soil respiration, which was also reflected by lower Reco regrowth rates in spring-summer seasons during the N$_2$O mitigation experiment (7.4 ± 1.3 g C m$^{-2}$ day$^{-1}$) than in years with normal management (8.2 ± 1.7 g C m$^{-2}$ day$^{-1}$).

Overall, being purely data-driven and highly site-dependent, the combination of machine learning models and SHAP

analysis confirmed the highly dynamic nature of the drivers influencing GPP and Reco and also revealed the complexity of their individual contributions across temporal scales. Compared to traditional linear models, these new approaches not only captured the non-linear and combined effects among drivers at high temporal resolution (here daily), but also provided explanatory outputs which enhanced interpretability and further improved our understanding of these complex grassland systems.

**4.3 Drivers during extreme growing seasons**

Across all extreme periods, SWC strongly reduced GPP (SHAP analysis 2; Fig. 6d). In contrast, the impact of VPD was small (Fig. 6d) even on days with exceptionally high VPD values (Fig. 6c). This finding supported our long-term driver analysis (SHAP analysis 1; Fig. 4), where SWC had a larger impact than VPD (mean absolute SHAP value of 0.6 for SWC and 0.2 for VPD during the peak growing seasons in 2018, 2019, 2022, and 2023), underscoring that soil water availability

was more relevant for GPP than atmospheric water demand. This observation aligns well with previous studies reporting more significant negative effects of droughts on grassland CO$_2$ fluxes compared to those of heatwaves (Li et al., 2016, 2021). It is also consistent with the behaviour seen during the 2003 summer drought (Teuling et al., 2010) and the 2018 summer





drought (Gharun et al., 2020), during which grasslands sustained evapotranspiration (ET) and thus GPP, as long as water was available in the soil, much in contrast to forests which rapidly reduced stomatal conductance and ET in response to high
VPD.

The effects of fertilizer N inputs were more differentiated during these extreme periods compared to the overall 20-year time series. While N inputs showed mostly negative impacts on GPP during the years 2018 and 2019 (i.e., during the $N_2O$ mitigation experiment and before the second renewal in 2021), effects of N inputs were typically positive during the years 2022 and 2023 (Fig. 6d). This suggests that after the second sward renewal in 2021, a resumed organic fertilization regime
with a higher amount of fertilizer N input could further help the recovery and regrowth after extreme periods (Hofer et al., 2016). Additionally, in response to the persistent droughts in 2018 and 2019, the farmer delayed or skipped some mowing events (which typically occurred monthly during the peak growing season, Fig. 6d and A2). In contrast, we observed more frequent mowing events (three times between June and August) during 2022 and 2023, which resulted in more regrowth periods and more variation in GPP compared to earlier years. This indicates that despite the frequent and more intense
extreme conditions in these two years compared to earlier years, the new grassland sward performed well and sustained the forage production during extreme summers.

Over the past two decades, Switzerland has experienced its hottest years on record (2022, 2023) and several extreme summers (e.g., 2018, 2022) (MeteoSwiss, 2023, 2024; Scherrer et al., 2022). In our study, we also observed consistent trends of rising temperature and frequent extreme summer months in recent years. The $CO_2$ fluxes of this intensively managed
grassland responded highly dynamically to changes in meteorology and already climate-adjusted management practices implemented by the farmer. However, it is unclear whether such small adjustments will suffice when extreme events intensify in the future (CH2018, 2018; IPCC, 2023). Although some evidence suggested that less intensive management may buffer grasslands against extreme impact (Winck et al., 2025), adopting a less intensive management scheme – which would require a different sward composition, livestock density, and adjusted management practices – is challenging for a single
grassland site in a short period. Therefore, site-specific adaptation of management strategies is needed to mitigate current and future climate risks (Gilgen and Buchmann, 2009; Woo et al., 2022). Further management options, such as multi-species mixtures (Craven et al., 2016; Finn et al., 2018; Isbell et al., 2015; Wang et al., 2025b) and precision-farming (Finger et al., 2019), can offer additional highly site-specific solutions to maintain selected ecosystem services and support the resilience of grasslands against extreme events.

**5. Conclusions**

Using a long-term dataset of $CO_2$ fluxes and detailed management information, our study provided unique and novel insights into temporal dynamics of $CO_2$ fluxes and their drivers. With state-of-the-art machine learning tools, we were able to identify and attribute the effects of different drivers, i.e., management and meteorological conditions, and their complex temporal dynamics, which would have been impossible with classical statistical approaches. While management-related



drivers showed high importance for grassland $CO_2$ fluxes, such information is unfortunately often lacking, creating large uncertainties in C cycle models at all scales. Therefore, we call for reliable collection and sharing of management data according to FAIR (Findable, Accessible, Interoperable, and Reusable) principles, alongside the highly valuable long-term observational data. Open science strongly improves our understanding of these highly dynamic grassland systems and enables development of realistic solutions under future climate conditions. Moreover, during the two decades of our

measurements, the grassland farmers succeeded in managing the site with relatively stable regrowth rates through adaptive management and interventions, already demonstrating effective climate-smart strategies. With increasing climate risks, additional practices like multi-species mixtures and precision farming, but also early warning systems for extreme events will become crucial to further enhance the resilience of grassland-based farming systems in the future.

**Appendix A**

**Table A1. The best hyperparameters used in the XGBoost models for GPP and Reco**

| Hyperparamters | GPP | Reco |
|---|---|---|
| n_estimators | 2000 | 2000 |
| max_depth | 5 | 5 |
| learning_rate | 0.05 | 0.1 |
| subsample | 0.8 | 1 |
| colsample_bytree | 1 | 1 |
| min_child_weight | 10 | 10 |
| reg_lambda | 10 | 10 |

**Table A2. GPP and Reco from 2005 to 2024 under three different scenarios: for the 16th (U_16), 50th (U_50), and 84th (U_84) percentile of the USTAR threshold distribution respectively.**

| Year | GPP_U50 | Reco_U50 | GPP_U16 | GPP_U50 | GPP_U84 | Reco_U16 | Reco_U50 | Reco_U84 |
|---|---|---|---|---|---|---|---|---|
| | Annual sum (g C m$^{-2}$ yr$^{-1}$) | | Mean and standard deviation of daily sum (g C m$^{-2}$ day$^{-1}$) | | | | | |
| 2005 | 1929 | 1646 | 5.1 ± 3.7 | 5.3 ± 4.0 | 5.7 ± 4.3 | 4.3 ± 2.0 | 4.5 ± 2.2 | 5.1 ± 3.0 |
| 2006 | 2059 | 1934 | 5.6 ± 4.2 | 5.6 ± 4.3 | 6.0 ± 4.4 | 5.1 ± 3.2 | 5.3 ± 3.3 | 5.4 ± 3.5 |
| 2007 | 2506 | 2270 | 6.9 ± 4.1 | 6.9 ± 4.0 | 7.2 ± 4.3 | 6.2 ± 3.5 | 6.2 ± 3.6 | 6.4 ± 3.8 |
| 2008 | 2764 | 2618 | 7.3 ± 4.5 | 7.6 ± 4.7 | 7.6 ± 4.6 | 6.6 ± 3.3 | 7.2 ± 3.6 | 7.0 ± 3.5 |
| 2009 | 2487 | 2200 | 6.9 ± 4.5 | 6.8 ± 4.3 | 7.1 ± 4.5 | 6.2 ± 3.4 | 6.0 ± 3.0 | 6.4 ± 3.3 |
| 2010 | 2055 | 1908 | 5.4 ± 3.5 | 5.6 ± 3.6 | 5.5 ± 3.6 | 4.9 ± 2.7 | 5.2 ± 2.9 | 5.0 ± 2.6 |
| 2011 | 2164 | 2004 | 5.7 ± 3.4 | 5.9 ± 3.5 | 6.1 ± 3.7 | 5.1 ± 2.4 | 5.5 ± 2.7 | 5.6 ± 2.8 |
| 2012 | 1483 | 1621 | 4.0 ± 3.7 | 4.1 ± 3.7 | 4.2 ± 3.8 | 4.3 ± 2.6 | 4.4 ± 2.7 | 4.5 ± 2.8 |



| Year | | | | | | | | |
|---|---|---|---|---|---|---|---|---|
| 2013 | 2036 | 1569 | 5.4 ± 3.8 | 5.6 ± 3.8 | 6.0 ± 4.2 | 4.2 ± 2.5 | 4.3 ± 2.6 | 5.0 ± 3.1 |
| 2014 | 2262 | 1759 | 6.3 ± 4.0 | 6.2 ± 3.9 | 6.7 ± 4.3 | 5.1 ± 2.6 | 4.8 ± 2.5 | 5.5 ± 2.9 |
| 2015 | 2112 | 1637 | 5.7 ± 3.5 | 5.8 ± 3.5 | 6.4 ± 3.8 | 4.4 ± 2.4 | 4.5 ± 2.5 | 5.4 ± 2.7 |
| 2016 | 2263 | 1781 | 6.1 ± 3.6 | 6.2 ± 3.6 | 6.6 ± 3.9 | 4.7 ± 2.2 | 4.9 ± 2.3 | 5.2 ± 2.5 |
| 2017 | 2429 | 2176 | 6.6 ± 4.3 | 6.7 ± 4.3 | 6.9 ± 4.4 | 5.8 ± 3.3 | 6.0 ± 3.3 | 6.1 ± 3.5 |
| 2018 | 1944 | 1898 | 5.2 ± 2.9 | 5.3 ± 3.0 | 5.8 ± 3.4 | 5.0 ± 2.2 | 5.2 ± 2.3 | 5.7 ± 2.8 |
| 2019 | 2123 | 2001 | 5.7 ± 3.1 | 5.8 ± 3.2 | 6.2 ± 3.4 | 5.3 ± 2.4 | 5.5 ± 2.5 | 5.9 ± 2.9 |
| 2020 | 2217 | 1969 | 6.0 ± 3.3 | 6.1 ± 3.3 | 6.4 ± 3.5 | 5.3 ± 2.3 | 5.4 ± 2.4 | 5.7 ± 2.6 |
| 2021 | 2061 | 1897 | 5.5 ± 3.8 | 5.6 ± 3.9 | 5.9 ± 4.1 | 5.0 ± 2.5 | 5.2 ± 2.7 | 5.4 ± 2.9 |
| 2022 | 2451 | 1970 | 6.8 ± 4.1 | 6.7 ± 4.0 | 7.3 ± 4.4 | 5.6 ± 2.8 | 5.4 ± 2.8 | 6.1 ± 3.0 |
| 2023 | 2274 | 1865 | 6.1 ± 4.1 | 6.2 ± 4.1 | 6.6 ± 4.4 | 4.9 ± 2.7 | 5.1 ± 2.7 | 5.5 ± 3.1 |
| 2024 | 2248 | 1950 | 6.1 ± 4.0 | 6.1 ± 3.9 | 6.4 ± 4.2 | 5.3 ± 2.8 | 5.3 ± 2.7 | 5.7 ± 2.8 |

**Table A3. Management events per parcel from 2021 to 2024, continuing Table B1 in Feigenwinter et al. (2023a). The number of each management activity in each year is indicated in round brackets. For grazing, the duration in days is given. Fertilizer N and C import, as well as total yields (from mowing and grazing) are given. Years where missing yield data are indicated (* yield from one mowing event missing; ** yield from one grazing event missing).**

| Year | Parcel | Management | Fertilizer N import (kg N ha⁻¹) | Fertilizer C import (kg C ha⁻¹) | Yield (kg DM ha⁻¹) |
|---|---|---|---|---|---|
| 2021 | A | Mowing (4), Grazing (6 d), Fertilization (4), Herbicide (1), Tillage (1), Resowing (2) | 237 | 1012 | 10127 |
| 2021 | B | Mowing (4), Grazing (6 d), Fertilization (4), Herbicide (1), Tillage (1), Resowing (1) | 198 | 795 | 9010 |
| 2022 | A | Mowing (5), Grazing (5 d), Fertilization (6) | 396 | 2384 | 13253* |
| 2022 | B | Mowing (5), Grazing (5 d), Fertilization (6) | 396 | 2384 | 13613* |
| 2023 | A | Mowing (6), Fertilization (5), Herbicide (1) | 291 | 1548 | 14065 |
| 2023 | B | Mowing (6), Fertilization (5), Herbicide (1) | 291 | 1548 | 13679 |
| 2024 | A | Mowing (4), Grazing (16 d), Fertilization (5), Herbicide (1), Harrowing (1), Rolling (1) | 314 | 1928 | 12527** |
| 2024 | B | Mowing (4), Grazing (16 d), Fertilization (5), Herbicide (1), Harrowing (1), Rolling (1) | 314 | 1928 | 12571** |




Table A4: Detailed information about management events from 2021 to 2024, continuing Table B2 in Feigenwinter et al. (2023a). The amount of harvested yield (kg DM ha⁻¹), fertilizer applied (m³ ha⁻¹), number of animals (number ha⁻¹) and duration (days) for grazing, as well as fertilizer N and C import (kg C or N ha⁻¹) are reported.

| Date | Parcel | Management | Amount (ha⁻¹)/Duration | Fertilizer C (kg C ha⁻¹) | Fertilizer N (kg N ha⁻¹) |
|---|---|---|---|---|---|
| 2021-02-23 | A, B | Organic fertilizer | 60 m³ ha⁻¹ A, 40 m³ ha⁻¹ B | 652 A, 435 B | 116 A, 77 B |
| 2021-05-03 | A, B | Mowing | 3712 kg DM ha⁻¹ A, 2452 kg DM ha⁻¹ B | | |
| 2021-05-07 | A, B | Organic fertilizer | 43 m³ ha⁻¹ | 138 | 50 |
| 2021-06-12 | A, B | Mowing | 2842 kg DM ha⁻¹ A, 2817 kg DM ha⁻¹ B | | |
| 2021-06-18 | A, B | Organic fertilizer | 28 m³ ha⁻¹ | 107 | 53 |
| 2021-07-21 | A, B | Mowing | 1919 kg DM ha⁻¹ A, 2548 kg DM ha⁻¹ B | | |
| 2021-07-29 | A, B | Organic fertilizer | 28 m³ ha⁻¹ | 115 | 18 |
| 2021-08-13 | A, B | Herbicide | | | |
| 2021-08-20 | A, B | Tillage (3-4 cm) | | | |
| 2021-08-20 | A, B | Resowing | 35 kg ha⁻¹ | | |
| 2021-09-09 | A | Resowing | 35 kg ha⁻¹ | | |
| 2021-10-14 | A, B | Mowing | 1126 kg DM ha⁻¹ A, 681 kg DM ha⁻¹ B | | |
| 2021-12-09 | A, B | Grazing | 529 kg DM ha⁻¹ A, 513 kg DM ha⁻¹ B, 36 Animals ha⁻¹, 6 days | | |
| 2022-03-14 | A, B | Organic fertilizer | 31 m³ ha⁻¹ | 516 | 83 |
| 2022-05-10 | A, B | Mowing | 5265 kg DM ha⁻¹ A, 5051 kg DM ha⁻¹ B | | |
| 2022-05-23 | A, B | Organic fertilizer | 26 m³ ha⁻¹ | 279 | 57 |
| 2022-06-10 | A, B | Mowing | Yield data not available. | | |
| 2022-06-27 | A, B | Organic fertilizer | 33 m³ ha⁻¹ | 542 | 78 |
| 2022-07-06 | A, B | Mowing | 2185 kg DM ha⁻¹ A, 2176 kg DM ha⁻¹ B | | |
| 2022-07-20 | A, B | Organic fertilizer | 29 m³ ha⁻¹ | 238 | 43 |
| 2022-08-09 | A, B | Mowing | 3160 kg DM ha⁻¹ A, 3322 kg DM ha⁻¹ B | | |
| 2022-08-25 | A, B | Organic fertilizer | 28 m³ ha⁻¹ | 293 | 56 |
| 2022-09-21 | A, B | Mowing | 1672 kg DM ha⁻¹ A, 1867 kg DM ha⁻¹ B | | |
| 2022-10-05 | A, B | Organic fertilizer | 43 m³ ha⁻¹ | 515 | 79 |
| 2022-12-11 | A, B | Grazing | 972 kg DM ha⁻¹ A, 1196 kg DM ha⁻¹ B, 45 Animals ha⁻¹, 5 days | | |
| 2023-03-21 | A, B | Organic fertilizer | 33 m³ ha⁻¹ | 261 | 60 |
| 2023-03-30 | A, B | Herbicide | | | |
| 2023-05-04 | A, B | Mowing | 3935 kg DM ha⁻¹ A, 4798 kg DM ha⁻¹ B | | |
| 2023-05-25 | A, B | Organic fertilizer | 26 m³ ha⁻¹ | 356 | 64 |
| 2023-06-11 | A, B | Mowing | 2627 kg DM ha⁻¹ A, 2559 kg DM ha⁻¹ B | | |
| 2023-06-19 | A, B | Organic fertilizer | 23 m³ ha⁻¹ | 288 | 53 |
| 2023-07-13 | A, B | Mowing | 2169 kg DM ha⁻¹ A, 1729 kg DM ha⁻¹ B | | |
| 2023-08-09 | A, B | Mowing | 1536 kg DM ha⁻¹ A, 1551 kg DM ha⁻¹ B | | |
| 2023-08-24 | A, B | Organic fertilizer | 24 m³ ha⁻¹ | 340 | 59 |
| 2023-09-25 | A, B | Mowing | 2812 kg DM ha⁻¹ A, 2282 kg DM ha⁻¹ B | | |
| 2023-09-29 | A, B | Organic fertilizer | 24 m³ ha⁻¹ | 302 | 55 |
| 2023-10-28 | A, B | Mowing | 985 kg DM ha⁻¹ A, 760 kg DM ha⁻¹ B | | |



| Date | Field | Activity | Details | | |
|---|---|---|---|---|---|
| 2024-01-20 | A, B | Grazing | Yield data not available. 33 Animals ha$^{-1}$, 4 days | | |
| 2024-02-28 | A, B | Organic fertilizer | 29 m$^3$ ha$^{-1}$ | 440 | 61 |
| 2024-03-19 | A | Herbicide | | | |
| 2024-03-22 | B | Herbicide | | | |
| 2024-04-11 | A, B | Mowing | 2387 kg DM ha$^{-1}$ A, 2914 kg DM ha$^{-1}$ B | | |
| 2024-05-02 | A, B | Organic fertilizer | 23 m$^3$ ha$^{-1}$ | 253 | 52 |
| 2024-06-11 | A, B | Mowing | 4119 kg DM ha$^{-1}$ A, 3827 kg DM ha$^{-1}$ B | | |
| 2024-06-20 | A, B | Harrowing | | | |
| 2024-06-20 | A, B | Rolling | | | |
| 2024-06-27 | A, B | Organic fertilizer | 30 m$^3$ ha$^{-1}$ | 454 | 67 |
| 2024-07-17 | A, B | Mowing | 2667 kg DM ha$^{-1}$ A, 2942 kg DM ha$^{-1}$ B | | |
| 2024-08-07 | A, B | Organic fertilizer | 25 m$^3$ ha$^{-1}$ | 370 | 59 |
| 2024-08-22 | A, B | Mowing | 1868 kg DM ha$^{-1}$ A, 1359 kg DM ha$^{-1}$ B | | |
| 2024-08-27 | A, B | Organic fertilizer | 31 m$^3$ ha$^{-1}$ | 412 | 75 |
| 2024-11-08 | A, B | Grazing | 1487 kg DM ha$^{-1}$ A, 1529 kg DM ha$^{-1}$ B, 65 Animals ha$^{-1}$, 12 days | | |


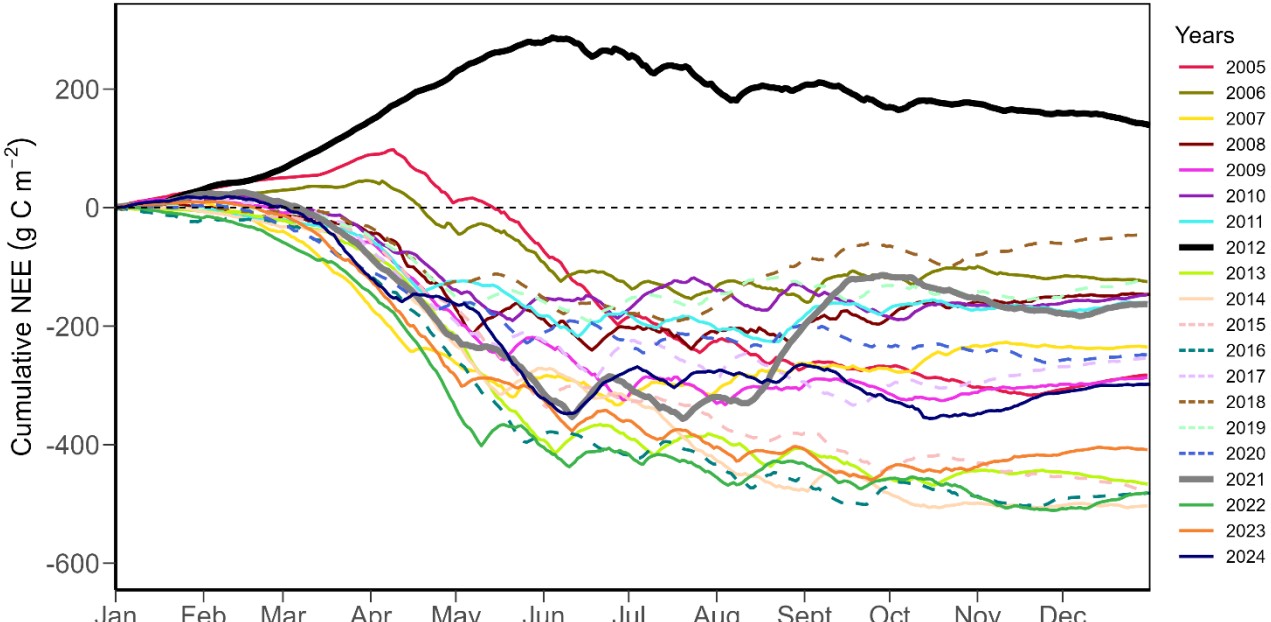

**Figure A1. Annual cumulative sum of daily net ecosystem CO₂ exchange (NEE) for the years 2005 to 2024. Years during the N₂O mitigation experiment (2015 to 2020; see Methods and Feigenwinter et al., 2023b) are shown with dashed lines, years with sward renewals (2012 and 2021) are shown in thicker black and grey lines, respectively.**



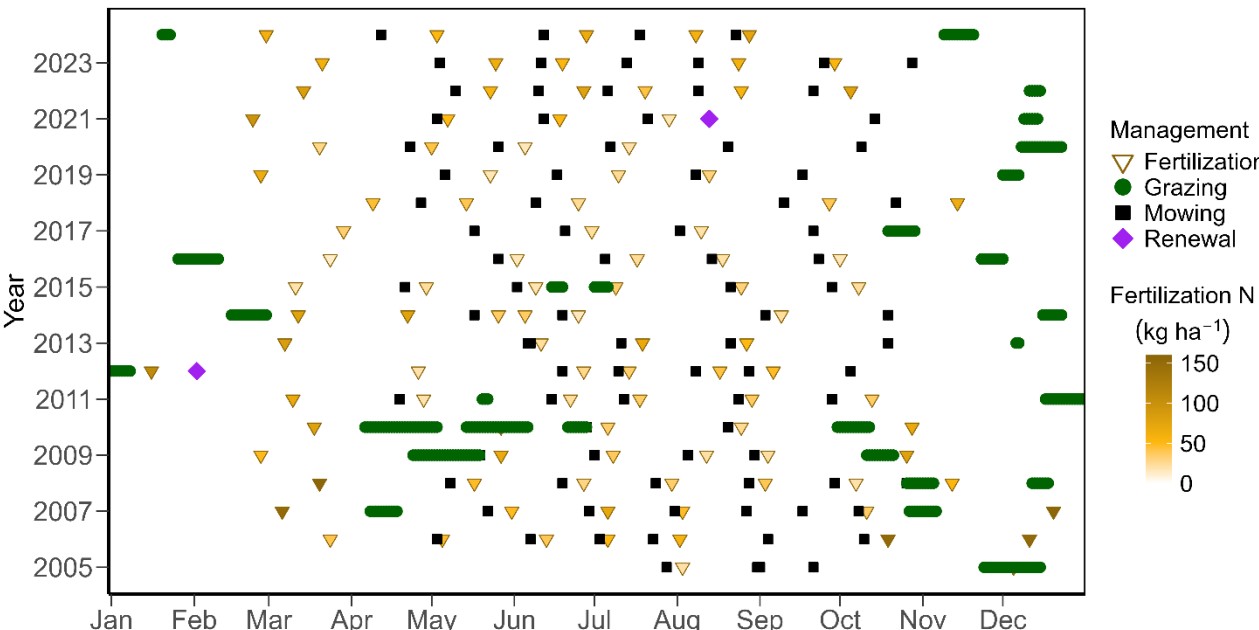


**Figure A2. Major management events at the Chamau grassland site between July 2005 and December 2024. The x-axis position of each point indicates the date of each event. Symbols represent different management activities: fertilization (brown triangles), mowing (black squares), grazing (green circles), and sward renewals (purple diamonds). Colours of fertilization triangles represent the amount of nitrogen inputs, with darker colours meaning higher inputs.**






**Figure A3. Performance and feature importance of XGBoost models for daily gross primary production (GPP) and ecosystem respiration (Reco). (a) Predicted versus actual GPP with model coefficient of determination ($R^2$) and root mean square error (RMSE) for the testing set, (b) feature importance (gain) for the GPP model, (c) predicted versus actual Reco with model $R^2$ and RMSE for the testing set, and (d) feature importance (gain) for the Reco model. In panels (a) and (c), red dashed lines represent**
**1:1 lines. In panels (b) and (d), higher gain values indicate higher importance of the features on model predictions. Features have the same colour code as in Figure 4: photosynthetic photon flux density (PPFD, red), soil temperature (TS, orange), air temperature (TA, yellow), soil water content (SWC, dark blue), vapour pressure deficit (VPD, blue), precipitation (PREC, light blue), day since last mowing, grazing or renewal (DaySinceUse, grey), daily fertilizer nitrogen input (DailyN, moss green), and gross primary production (GPP, green).**




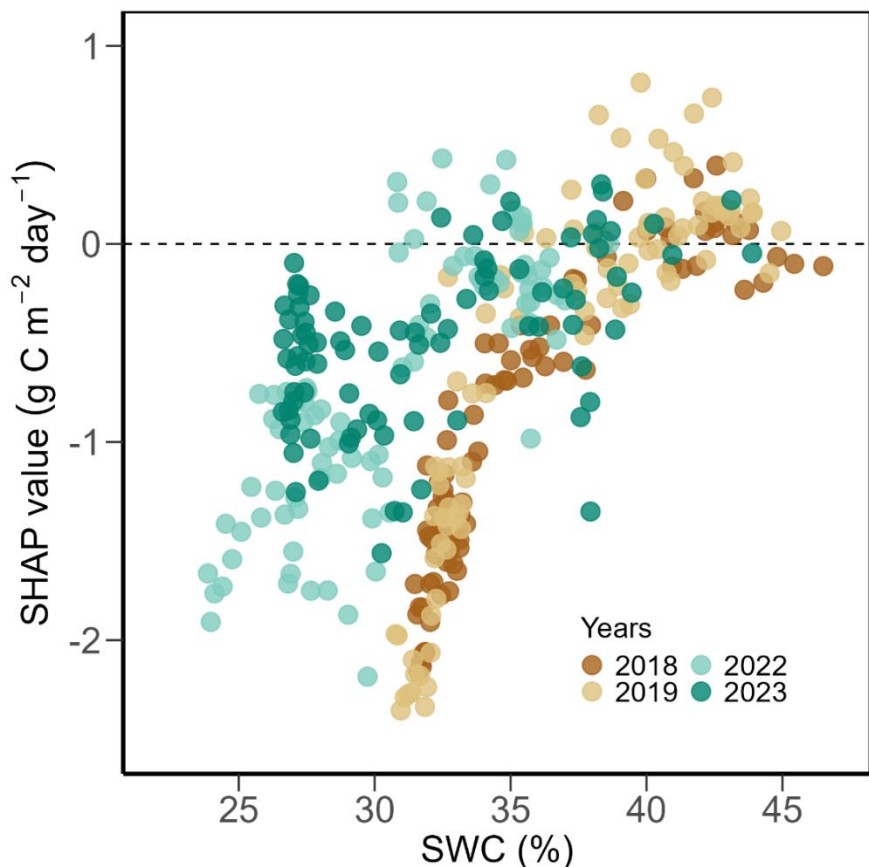


**Figure A4. Partial dependence plot of SHAP values of soil water content (SWC) for predicted daily gross primary production (GPP) in the peak growing season (June, July, August) during the years with extreme conditions 2018, 2019, 2022 and 2023 (see Fig. 2) from SHAP analysis 2 (Fig. 6). The mean prediction at SHAP value zero (dashed line) corresponds to a GPP of 9.7 g C m$^{-2}$ day$^{-1}$. Positive SHAP values represent positive impacts of SWC on GPP relative to the mean prediction, while negative values**
**indicate negative impacts from the mean prediction. Different colours represent different years.**

**Code availability**

The R and Python scripts used for the data analyses and visualizations are available upon request from the corresponding author. All scripts used in producing the PI dataset are available on GitHub (https://github.com/holukas/dataset_ch-cha_flux_product).

**Data availability**

Data used in this study will be openly available for download in the ETH Zurich Research Collection at https://doi.org/10.3929/ethz-b-000745429 (Wang et al., 2025a; Preliminary link).



**Author contribution**

YW, IF, and NB: conceptualization of the study. YW, IF, and LH: data curation. YW: formal analysis, visualization, and
writing (original draft preparation). IF and NB: supervision. AKG and NB: project administration. NB: funding acquisition.
All authors: methodology and writing (reviewing and editing).

**Competing interests**

The authors declare that they have no conflict of interest.

**Acknowledgements**

The technical assistance for the maintenance of the eddy covariance station by Peter Plüss, Thomas Baur, Philip Meier,
Patrick Koller, Florian Käslin, Paul Linwood, Markus Staudinger, Peter Ravelhofer, and Martin Rüegg is greatly
acknowledged. Big thanks also go to all the scientists who were responsible for the Chamau site during the 20 years of this
study: Matthias Zeeman, Werner Eugster, Lutz Merbold, Dennis Imer, and Kathrin Fuchs. We also thank Lukas Stocker and
the staff from LBBZ Schluechthof team for managing the fields around the flux station. We acknowledge fruitful discussions
with Wei Qiu, University of Washinton, Seattle, USA about machine learning models, in particular on SHAP analyses. AI
tools (ChatGPT-4o, Grammarly, and DeepL) were used to improve writing and visualization. All outputs were critically
reviewed by the authors to ensure accuracy and integrity.

**Financial support**

We acknowledge funding for the SNF projects InsuranceGrass (100018L_200918) and ICOS-CH Phase 3 (20FI20_198227).
NB and YW are part of the SPEED2ZERO, a Joint Initiative co-financed by the ETH Board.

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
