# Peer review of "Drivers of long-term grassland CO2 fluxes and regrowth: effects of management and meteorological conditions over time"

_EGUsphere, 2025_

## Referee Comment (RC3)

Grasslands constitute a significant portion of agricultural landscapes in Europe and store substantial amounts of carbon. Effective management of these ecosystems serves as a nature-based solution for enhancing carbon sequestration. This study presents a unique long-term dataset spanning 20 years, encompassing flux, biometeorological, and detailed management data from an eddy covariance site in a Swiss grassland. It addresses the gap in understanding the temporal dynamics and development of $CO_2$ drivers by employing machine learning models, as opposed to the predominantly linear models used previously.

This research contributes to a detailed understanding of the impact of various management practices on the carbon budgets of mid-latitude grasslands. The use of state-of-the-art machine learning models provides an extended understanding of non-linear and dynamic relationships between $CO_2$ flux and its drivers. The study is methodologically sound, with comprehensive descriptions of the eddy covariance setup, the data processing, and statistical analyses. However, some concepts require further clarification. The results are well discussed, figures are well-described and visually comprehensible.

**Specific comments:**

*Methods & Results*:

1. Line 147: the soil variables are averaged across depths - what is the reasoning behind that? Especially for the soil water content measurements were taken in very different depths, and soil water fulfill different functions in the top soil layer than in the deep soil layer, depending on how available it is to microbes or plant roots-> why not use separately and see which layer is important in which times to gain more process understanding?

2. Line 114-115: DailyN is calculated as a daily average across the regrowth period. However, not the same amount of N would be available to the plants each day, maybe in the beginning more and later less, depending also on environmental factors such as strong rain events and leaching – please elaborate in how far this has implications on your results.

3. Line 140: The gaps in flux data were filled with a random forest model. Might these confound the driver analysis that is conducted later? E.g. if gap-filling is based on other environmental variables such as Tair or VPD, of course the driver analysis shows high dependence of ecosystem fluxes on these variables. Please clarify, and if there might be confounding effects elaborate on that in the discussion.

4. Figure 4:
   For the partitioning of GPP and Reco the nighttime partitioning method by Reichstein et al (2005) was used. In this method, This means, Reco is modelled depending on temperature. GPP is calculated as the difference between measured NEE and modelled Reco. Of course there will be dependencies then of the component fluxes to temperature and GPP, and their

contribution in explaining the fluxes might be partly conceptually introduced. Does it "even out" since you integrate over regrowth periods?

I would suggest to quickly test another partitioning method (e.g. hybrid partitioning method introduced by Nguyen et al (2025), code available on github) to confirm if the main drivers are the same or if the results are confounded the partitioning method.

Nguyen, N. B., Migliavacca, M., Bassiouni, M., Baldocchi, D. D., Gherardi, L. A., Green, J. K., Papale, D., Reichstein, M., Cohrs, K.-H., Cescatti, A., Nguyen, T. D., Nguyen, H. H., Nguyen, Q. M., and Keenan, T. F.: Widespread underestimation of rain-induced soil carbon emissions from global drylands, Nat. Geosci., https://doi.org/10.1038/s41561-025-01754-9, 2025.

**Technical corrections:**

Line 95: how many times per year/ in which times fertilizer is applied? Please specify. One fertilizer application per regrowth period?

Line 128: "For more detailed information on instrumentation, see ⋯"

Line 205: mean annual temperature and precipitation is a repetition from lines 91-92, can be removed here or just put in brackets, e.g. "mild temperate climate (9.9 °C, 1147 mm)"

Line 354-356: the two parts of the sentences are not necessarily connected, I would suggest to split it up into two sentences. First part ("Often, studies focused on natural ecosystems or
355 forests at regional and global scales (Anav et al., 2015; Cai and Prentice, 2020; Davi et al., 2006; Norby et al., 2010)") fits better in the Introduction.

Line 380-383: please improve for more clear language: "Such observations, albeit rare, are impactful in terms of C dynamics, and underscore the necessity to include these infrequent destructive management practices in long-term flux studies  as well as modelling frameworks."

Line 392-393: sentence is unnecessarily long and not so well understandable, please split it up into two shorter sentences

Line 428: what do you mean with "explanatory outputs which enhanced interpretability"? It sounds a bit generic, please specify

Line 436: in line 431 you state that during extreme events "SWC strongly reduced GPP", but later you state that "more significant negative effects of droughts on grassland $CO_2$ fluxes compared to those of heatwaves". It seems a bit contradictory, maybe you can clarify this section.

Line 457-460: complicated sentence structure, please simplify or split into two sentences

Line 462: please mention briefly what precision-farming is (and please mind consistent terms – in L. 477 precision farming is written without "-")

---

## Author Comment (AC1)

**Author Comments – Response to Referee 1**

Referee comments are marked in black and author responses are marked in blue.

General comments:
This is a well-written paper that reports on 2 decades of $CO_2$ exchange measurements at an intensively managed grassland in Switzerland with the aim of disentangling the influence of management amidst variable environmental conditions and ongoing climate change. The paper builds up on Feigenwinter et al. (2023) who analyzed the first 16 years, here the focus is more on the analysis of the drivers using a machine learning approach. I think the manuscript is ready for publication following some minor changes, detailed below.

Dear Dr. Wohlfahrt,
Thank you for your positive comments! We appreciate the opportunity to improve the manuscript based on your constructive feedback. We have addressed your comments and suggestions in the responses below.

There are however two terminology issues that I struggle with and ask the authors to consider:

- The authors analyze what they refer to as regrowth periods in between management events, especially harvesting. What I struggle with is the terminology "GPP/RECO regrowth rates" which the authors use to refer to the GPP/RECO during the regrowth periods. The terminology to me however suggests GPP/RECO "to regrow", i.e. rebound, during these periods, which may not be the case. In fact, the negative SHAP values for days since last management and GPP suggest a negative relationship. I think the authors could simply say something like GPP/RECO during regrowth periods, which may be a little awkward at times, but less ambiguous.

Thank you for pointing out this ambiguity. We will change the term to "GPP or Reco during regrowth periods" throughout the text. For each regrowth period, cumulative GPP and Reco were first calculated and then averaged based on the length of the regrowth period.

Moreover, taken together with the comments from Referee 2, to put more emphasis on "regrowth periods" instead of just "regrowth", we will also modify the title of the manuscript as "Drivers of long-term grassland $CO_2$ fluxes: effects of management and meteorological conditions during regrowth periods".

- The authors suggest, e.g. in the abstract but also elsewhere, that the fact that there was no trend in $CO_2$ exchange over the two decades despite ongoing climate change shows that the farmers are using a climate-smart management. This statement to me implies that the management is deemed climate-smart as it prevented a decrease in the $CO_2$ sink strength. This ignores the possibility that an alternative (truly climate-smart?) management could have profited from ongoing climate change and increased the sink strength. Neither option (a decrease or increase in sink strength was prevented by the actual management) can be answered with the present data that are conditional on the actual management. This would need a manipulative experiment (with alternative management like in Ammann et al. 2007) or the use of some model which represents management and the resulting consequences on $CO_2$ fluxes (which would be an intriguing follow-up). I thus suggest to down-tune the climate-smart aspect and rather leave it with saying that the adaptive management that the farmer practiced in response to interannual and intra-seasonal variability in weather conditions apparently was able

to keep $CO_2$ exchange stable in the face of ongoing climate change during the two decades of observations.

We appreciate this input. 'Climate-smart' agriculture (CSA) includes a set of practices and technologies that improve productivity, while enhancing resilience and reducing GHG emissions during on-going climate change (FAO, 2019; World Bank, 2024). The FAO defines CSA as "an approach that helps guide actions to transform agri-food systems towards green and climate resilient practices". Similarly, the World Bank defines CSA as "an integrated approach to managing landscapes—cropland, livestock, forests and fisheries—that address the interlinked challenges of food security and climate change.". Science communities have been using this term as well since long (e.g., Lipper et al., 2014; Walter et al., 2017). We will include this context in the introduction.

With more frequent extreme events in recent years that were observed in our time series, we would expect a decreasing trend in GPP during regrowth periods and ultimately a decrease in $CO_2$ sink strength. In contrast, the non-significant trend detected in our GPP data shows that the existing management practices were able to maintain productivity, thus suggesting resilience to extreme events which is considered 'climate-smart'. Meanwhile, we also agree that more "climate-smart" management practices aiming to improve resilience and sustainability or even increase productivity of agroecosystems under ongoing climate change – albeit in the absence of extreme events, have to be tested with experiments or certain models, for example to see the effect of the timing and intensity of certain management practices on productivity and GHG emissions. We indeed have ongoing work in the group using the process-based model MONICA (Nendel et al., 2011) on this exact topic (Kamali et al., submitted). With all these considerations in mind, we will put our argumentation in context and explain this aspect better throughout the manuscript.

Detailed comments:
1. l. 8: the temporal development of management practices and meteorological conditions is uncertain? Aren't the interactive effects of these on grassland $CO_2$ fluxes uncertain?

We argue that both the temporal development of the drivers themselves and their impact/effects are uncertain. We will rephase this sentence into "… $CO_2$ fluxes of managed grasslands are substantially influenced by land management practices and meteorological conditions, but the temporal development of drivers and their effects are still uncertain".

2. l. 9-10: this sentence could be removed in the abstract without loss of information

We wanted to introduce the terms of NEE, GPP, and Reco in the very beginning, but we also introduced these terms again in the introduction. We will delete this sentence as suggested and introduce the abbreviations when they are first mentioned in the abstract.

3. l. 42: GPP and RECO are the essential part of C cycling of any ecosystem

Agreed. We will change the sentence into "As an essential part of ecosystem C cycles…".

4. l. 71: in my view Wohlfahrt et al. (2008, 10.1029/2007JD009286) were one of the first grassland papers to look into the interactive effects of management and environmental drivers and in fact also analyzed data in periods stratified by management (harvesting) events

Thank you for the comment. We were aware of this study and cited it later in the discussion. Here we focused on "long-term studies", but we will mention this 6-year study in the introduction as well.

5. l. 130: what about the self-heating correction of the Li-7500 – I guess at least during the early phase of the time series the used models required this correction? In addition, the early Li-7500 models had some intrinsic lag of the digital signals that could be increased on the software side to result in a lag that is some multiple of the sampling rate in order to be removed – what lag value was set – 0.3 s?

Regarding self-heating: No self-heating correction was applied to open-path LI-7500 fluxes. There are several reasons for this decision:

(1) We found that the current standard self-heating correction (Burba et al., 2008) produced unsatisfactory and unreliable results at multiple Swiss FluxNet sites. Comparative analysis using parallel (en-)closed path measurements (LI-7200 vs. LI-7500) at these sites revealed significant, conflicting biases depending on the dataset (e.g., strong underestimation or overestimation of NEE, as detailed in Figures R1-1 and R1-2, respectively). Our observations, demonstrating that the standard correction can lead to fluxes substantially deviating from the "true" flux, are consistent with similar findings by Wohlfahrt et al. (2008, their Fig. 3), who utilized an earlier correction version (Burba et al., 2006). Furthermore, we note that the Burba et al. (2008) correction was derived from a limited dataset, validated specifically for vertically mounted IRGAs, and does not account for the non-vertical (15° tilted) installation geometry of the LI-7500 at our site.

[Figure]

**Figure R1-1**. Cumulative NEE fluxes from Feb 2015 until Apr 2017 at the cropland site CH-OE2. Shown is a comparison of self-heating correction approaches: open-path LI-7500 fluxes (only WPL corrected, red) with enclosed-path LI-7200 fluxes (black, assumed to show the "true" flux). Also shown are cumulative fluxes after applying the Burba et al. (2008) correction as implemented in EddyPro using the single linear regression method (SLR, blue) and the multiple regression method (MLR, purple).

[Figure]

**Figure R1-2**. Cumulative NEE between Jan 2016 and Dec 2017 at the forest site CH-LAE. "True" flux from LI-7200 (black), no self-heating correction for LI-7500 fluxes (red), after correction in EddyPro Burba 2008 (blue) and after correction Kittler 2017 (orange).

(2) The absence of concurrent validation data from a co-located (en-)closed path IRGA, such as the LI-7200, introduces a significant methodological uncertainty when applying the self-heating correction. Without these parallel measurements, we cannot independently validate the corrected fluxes, creating an unverified "black box" scenario. This black box application of any available correction poses a substantial risk to data quality, a concern echoed by Deventer et al. (2021). They highlighted that utilizing current self-heating corrections without parallel reference flux measurements "[...] yields uncertainties that are larger than random flux errors - substantially degrading confidence in ecosystem carbon [...] budgets", underscoring the necessity of empirical validation.

(3) For forest sites in the Swiss FluxNet, we apply the correction described in Burba et al. (2006), with the modification that we also apply a scaling term ξ to account for the tilted angle of the LI-7500 (similar to Kittler et al., 2017; see orange line in Figure R1-2). The scaling term is site-specific and must therefore be determined empirically from parallel measurements. We have tried to generalize ξ, based on data from other non-forest sites (grassland, cropland), and found that ξ can be complex with variations over the course of one day and differences between daytime and nighttime data. We concluded that the correction is not possible without parallel measurements.

(4) We are cautiously optimistic that the self-heating effect at this site is small. In July 2025, we started parallel measurements at CH-CHA. We found a mean difference of approx. 3% between the LI-7500 and LI-7200 fluxes, with the open-path showing slightly more uptake (Figure R1-3). The main difference was found during a time period characterized by high temperatures >= 32°C at the end of July 2025. However, during the preceding weeks in July, NEE from the two IRGAs were virtually identical. We are aware that the correction was originally meant for colder climate conditions, in particular with air temperatures < -10°, but we currently have no winter LI-7200 data from CH-CHA. In a comparison of parallel measurements during winter for a high-altitude alpine grassland (CH-AWS, about 2000 m a.s..l.) with a comparable setup, we found that the self-heating effect was small, similar to Haslwanter et al. (2009). Parallel measurements at CH-CHA will continue, and we will investigate more data once available.

[Figure]

**Figure R1-3**. Cumulative, directly-measured (not gap-filled) NEE fluxes measured at the grassland CH-CHA in 2025. Fluxes were measured with an open-path LI-7500 (IRGA75, green) and (en-)closed path LI-7500 (IRGA72, blue).

We will add more details in the Methods on this aspect.

Regarding intrinsic lag: All raw data coming from the sonic and IRGA were directly logged using the custom made, real-time logging software *sonicread* (concept described in Eugster and Plüss, 2010), circumventing the LICOR software to store data. Found time lags for the LI-7500 $CO_2$ and $H_2O$ signals were between 0.20 s and 0.35s throughout all years. We detected the time lags on a yearly basis and collected results from a detailed analysis (results available online: https://holukas.github.io/dataset_ch-cha_flux_product/L0.html#openlag-runs-to-determine-final-lag-ranges).

6.  l. 135: which approach for flux partitioning was used – day or nighttime?

The nighttime partitioning method was used. We mentioned this in the next paragraph (line 143 in the original manuscript). We will add this info earlier in a revised version of the manuscript.

7.  Fig. 1e: given the length of the time series I feel a bit overwhelmed with the day-to-day variability and thus I suggest showing $CO_2$ fluxes on a monthly timescale, possibly as a stacked bar chart that might nicely visualize the interplay between GPP and RECO on NEE

Thank you for this comment. We presented daily fluxes to show the basis of our analysis. Since this site is being intensively managed, common aggregation methods (e.g., monthly mean/sum or weekly mean/sum) do not represent this complex situation and information about management will – in the best case – be lost, or – in the worst case – bias the results during any longer aggregation. Therefore, we prefer to keep Figure1e as is. In Figure 3, we actually already show GPP and Reco aggregated for on the regrowth periods.

8. Fig. 1: would it possible to add an additional panel that shows the cutting events, grazing periods and re-sowing events?

Thank you for the suggestion. The original Figure A2 was meant to present all management information in detail. Given the width of the figures and the frequency of the management events within each year, putting all these events as an additional, sixth panel would make the figure very crowded. However, we received further comments from colleagues supporting the wish to not "hide" this management info in the appendix, since such info is typically very rare in such detail. In a revised version of the manuscript, we will thus move Figure A2 to the main text as a panel in Figure 3 (see below, comment 10).

9. Table 1: is huge but conveys limited information and might thus go into the supplement?

Agreed. We will move this table to the appendix as Table A2.

10. Fig. 3: I suggest adding Fig. A2 as a third panel here; overall the information content of this figure is limited - GPP/RECO is smaller during the off-season period with short days and larger during the warm period with long days

Thank you for this comment. We will combine the original Figure A2 with the GPP and Reco panels as a new Figure 3 as shown below (also answering to comment 8 above).

[Figure]

**Figure R1-4**. New Figure 3 for the revised manuscript

11. l. 270: DaysSinceUse shows negative SHAP values for GPP – correct? Does that mean that GPP declines the more time has passed since the start of the regrowth period? If so, might be worthing spelling this out

Thank you for the question. Indeed, DaySinceUse shows negative SHAP values for GPP in Figure 4 (based on SHAP analysis 1). Since the bars presented in Figure 4 are based on average SHAP values from each regrowth period, they represent the average effect of this variable on GPP, always compared to the grand mean or mean prediction of all GPP for all 115 regrowth periods. On the daily scale (Figure R1-5 below), SHAP values for DaySinceUse are normally very negative directly after moving/grazing (DaySinceUse < around 10 days) and then increase. This correctly reflects the effect of mowing/grazing: the more days since use, the higher GPP since the grassland could regrow, i.e., the more days since use, the larger the SHAP value for this driver, compared to the mean prediction (as explained in the figure caption). To avoid such confusion, we will add the partial dependence plot of DaySinceUse and its SHAP value in the appendix as part of new Figure A3. We will also add a sentence in Section 3.1 at current line 271: "When averaged over the entire regrowth period, SHAP values of management events (i.e., mowing and grazing, represented by DaysSinceUse) on GPP were often negative (Fig. 4a). However, at daily scale (Fig. A3c), SHAP values for DaySinceUse were first negative and then steadily increased before staying stable after around 20 days, indicating that GPP increased the more time had passed since the last management event.".

[Figure]

**Figure R1-5**. SHAP dependence plot for DaySinceUse of GPP (zoomed in to the first 60 days after management)

12. l. 311: these significant differences are not visible from Fig. 6b-c

Thanks for pointing this out. This significant difference is more obvious in Figure 2. with higher VPD and lower SWC on a monthly scale. We will change the sentence into "Focusing on years with more extreme summer months (i.e., July and August of 2018, July of 2019, July of 2022, and June of 2023; Fig. 2), we found normal daily PPFD (Fig. 6a), but significantly higher-than-normal daily temperature and VPD, accompanied by lower-than-normal soil water content (Fig. 6bc). Mean monthly air temperature in July 2018, 2019, 2022 was 1.2, 1.2, and 1.1 °C above the 20-year monthly average (19.6 °C) respectively, while 1.8 °C higher in August 2018

compared to 20-year monthly average (18.6 °C). All extreme summer months had SWC more than 10% lower than the 20-year monthly averages.".

13. l. 372-377: these data should be first introduced in the Results section

Thank you for the suggestion. We mentioned these cumulative NEE numbers already in Section 3.1 (lines 230-233 in the original manuscript). We will add the numbers for renewal years to current line 232 as "… (Fig. A1). In 2012 and 2021 two grassland renewal events (i.e., ploughing and reseeding of the grassland) took place, with very different effects on annual NEE: while the event in 2012 led to a strong net $CO_2$ loss (cumulative NEE of 139 g C $m^{-2}$ $yr^{-1}$; Fig. A1), the event in 2021 resulted only in a weaker-than-normal net $CO_2$ uptake (cumulative NEE of -163 g C $m^{-2}$ $yr^{-1}$; Fig. A1). All three flux components…". We will reformulate the discussion at current line 372 as "… such events. The observed differences in annual NEE during the two renewal events at the Chamau grassland (139 vs. -163 g C $m^{-2}$ $yr^{-1}$ for 2012 and 2021, respectively; Fig. A1) might be explained by differences in seasons (February 2012 versus August 2021) and soil disturbance intensities (ploughed at 20 cm in 2012 and 3-4 cm in 2021), which subsequently influenced the establishment and regrowth of the new sward. Other …"

14. l. 471: … in C cycle model simulations …

Thank you for the comment. We will change the sentence to "… creating large uncertainties in C cycle model simulations at all scales".

**References**
Burba, G. G., Anderson, D. J., Xu, L., McDermitt, D. K.: Correcting apparent off-season $CO_2$ uptake due to surface heating of an open-path gas analyzer: progress report of an ongoing study. Proceedings of 27th Conference on Agricultural and Forest Meteorology, May 22–26, San Diego, CA, USA, 2006.

Burba, G. G., McDermitt, D. K., Grelle, A., Anderson, D. J., and Xu, L.: Addressing the influence of instrument surface heat exchange on the measurements of $CO_2$ flux from open-path gas analyzers, Global Change Biol., 14, 1854–1876, https://doi.org/10.1111/j.1365-2486.2008.01606.x, 2008.

Deventer, M. J., Roman, T., Bogoev, I., Kolka, R. K., Erickson, M., Lee, X., Baker, J. M., Millet, D. B., and Griffis, T. J.: Biases in open-path carbon dioxide flux measurements: Roles of instrument surface heat exchange and analyzer temperature sensitivity, Agric. For. Meteorol., 296, 108216, https://doi.org/10.1016/j.agrformet.2020.108216, 2021.

Eugster, W. and Plüss, P.: A fault-tolerant eddy covariance system for measuring $CH_4$ fluxes, Agric. For. Meteorol., 150, 841–851, https://doi.org/10.1016/j.agrformet.2009.12.008, 2010.

FAO: Climate-Smart Agriculture, https://www.fao.org/climate-smart-agriculture/en/, 2019.

Haslwanter, A., Hammerle, A., and Wohlfahrt, G.: Open-path vs. closed-path eddy covariance measurements of the net ecosystem carbon dioxide and water vapour exchange: A long-term perspective, Agric. For. Meteorol., 149, 291–302, https://doi.org/10.1016/j.agrformet.2008.08.011, 2009.

Kamali, B., Buchmann, N., Feigenwinter, I., Wang, Y., Ewert, F., Gaiser, T.: Navigating the trade-off among biomass production and GHG emissions for smart management of grasslands, Submitted.

Kittler, F., Eugster, W., Foken, T., Heimann, M., Kolle, O., and Göckede, M.: High-quality eddy-covariance $CO_2$ budgets under cold climate conditions, J. Geophys. Res.: Biogeosci., 122, 2064–2084, https://doi.org/10.1002/2017JG003830, 2017.

Lipper, L., Thornton, P., Campbell, B. M., Baedeker, T., Braimoh, A., Bwalya, M., Caron, P., Cattaneo, A., Garrity, D., Henry, K., Hottle, R., Jackson, L., Jarvis, A., Kossam, F., Mann, W., McCarthy, N., Meybeck, A., Neufeldt, H., Remington, T., Sen, P. T., Sessa, R., Shula, R., Tibu, A., and Torquebiau, E. F.: Climate-smart agriculture for food security, Nat. Clim. Change, 4, 1068–1072, https://doi.org/10.1038/nclimate2437, 2014.

Nendel, C., Berg, M., Kersebaum, K. C., Mirschel, W., Specka, X., Wegehenkel, M., Wenkel, K. O., and Wieland, R.: The MONICA model: Testing predictability for crop growth, soil moisture and nitrogen dynamics, Ecol. Modell., 222, 1614–1625, https://doi.org/10.1016/j.ecolmodel.2011.02.018, 2011.

Walter, A., Finger, R., Huber, R., and Buchmann, N.: Smart farming is key to developing sustainable agriculture, PNAS, 114, 6148–6150, https://doi.org/10.1073/pnas.1707462114, 2017.

Wohlfahrt, G., Fenstermaker, L. F., and Arnone Iii, J. A.: Large annual net ecosystem $CO_2$ uptake of a Mojave Desert ecosystem, Global Change Biol., 14, 1475–1487, https://doi.org/10.1111/j.1365-2486.2008.01593.x, 2008.

World Bank: Climate-Smart Agriculture, https://www.worldbank.org/en/topic/climate-smart-agriculture, 2024.

---

## Author Comment (AC2)

**Author Comments – Response to Referee 2**

Referee comments are marked in black and author responses are marked in blue.

Overall assessment (do not require specific comments by the authors)
The study presents and analyses a rare, high-quality long-term data set of field measurements in an intensively managed ecosystem, led by the colleagues who run the site. The data includes $CO_2$ exchange and management data. The scientific focus is on identifying drivers for interannual variability of gross-primary productivity (GPP) and ecosystem respiration (Reco) in 'normal' and extreme years. The manuscript clarifies nicely that management and especially the aboveground and the complete renewals of the grassland vegetation mark significant short and long-term disturbances putting additional constraints on comparability of periods and years. Weather and management variability and their interactions, which I believe can be expected by the concept of adaptive climate smart management, characterize the challenge, when investigating this ecosystem, compared to, e.g., natural vegetation.

The work creates order in the time series through some useful classification. A period of 20 years is long enough to identify significant environmental trends and weather extremes. The period includes two grassland renewal events and numerous aboveground canopy harvests that define regrowth periods (RPs), still of different length and located in differing seasons. The simplicity and clarity of these decisions to define the perspectives on how to look at the time series and analyse it, is one of the strengths of this work. One sub-period of 5 years, i.e. still ¼ of the investigated period, when part of the study area was subject to a comparative experiment, adds complexity and the way this has been dealt with, raises some questions.

The complexity of the data set requires a complex analysis approach with careful selection of drivers, which I will discuss below. Going for a machine learning approach (XGB) for analysis might be a good choice, as it puts the least constraints on the results compared to alternatives, such as alternative empirical mechanistic modelling approaches. But the different nature of the results, especially the results from the SHAP analysis, are yet a bit difficult to understand and I suggest more explanation and guidance for the reader.

In general, I see this work as a model for such long-term empirical studies in managed ecosystems, for sure a strong scientific contribution to quantitative Biogeosciences.

Dear Dr. Ibrom,
Thank you for your positive and constructive comments! We appreciate the opportunity to improve the manuscript based on your thorough review and feedback. We have addressed your comments and suggestions in the responses below.

Some general critical comments are as follows (please comment and take action, where applicable):

G1: Devising extreme months with a clear and simple Z-score approach on soil physical and atmospheric drivers for drought makes good sense. The classifying variables are well chosen, because a capacitive variable (SWC) and an atmospheric state variable (VPD) are combined to represent the accumulated (SWC deficit) and actual stress (VPD). For the latter, potential evaporation might possibly be an even stronger variable.

Thank you for the comment. There are different approaches to how to represent stress. Based on existing work (Liu et al., 2025; Novick et al., 2024; Shekhar et al., 2023), we think that VPD

and SWC are more representative to the actual stress of the ecosystem rather than potential evaporation. Furthermore, from a reader's perspective, VPD is more straightforward and easier to understand as a stress factor compared to potential evaporation.

G2: Machine learning methods are relatively novel, and I believe the interpretation of the results is still a challenge. From my own experience with this text, there is a large risk for a reader, not yet familiar with the SHAP analysis, of miss-interpreting the results from the SHAP analysis, e.g., a "negative effect" as "negative relationship" between a driver (D) and a response variable (R). A negative SHAP value ("negative effect") shows rather only that the contribution from D has made R lower than the reference. This is irrespective of the sign of a relationship: a positive relationship ($dR/dD > 0$) makes R small at low values of D a negative relationship ($dR/dD < 0$) makes R small at high values of D. To avoid misinterpretation by the readers, please consider explaining this possible trap for understanding the text correctly.

We appreciate this constructive input. The explanation of SHAP values needs to consider the value of the feature/driver for each single observation as well, and indeed 'effect' is the appropriate way to describe it. The sign of the SHAP value also depends on the response variable, for example, if NEE is the response variable, negative SHAP value would mean that this driver increases the $CO_2$ uptake. We are aware that SHAP analysis is still relatively new to the community and readers. To avoid any misinterpretation of the SHAP values, we had already added detailed explanations in the method section and the respective figure captions in the original manuscript. Moreover, we never talked about a negative 'relationship' in the text. Since we see more and more SHAP analyses in the scientific literature (Krebs et al., 2025; Li et al., 2025), we think that our explanations should suffice. In any case, we will carefully check the wording throughout the text to make sure 'effect' is used where appropriate.

G3: However, I claim, for scientific understanding, relationships are more relevant than just effects. By examining the effects further, e.g. by looking into the relationship between the effect and the magnitude of D, you might be able to say something more about the nature of the relationship (see, e.g., D29) and possible interactions between drivers.

Thank you for the comment. We fully agree that 'effects' and 'relationships' are different, and it is important to explain both. In machine learning models, looking at only bilateral relationships between one driver and the response variable cannot be achieved, as in conventional linear additive statistics, the number of drivers and their interactions are strongly limited. PCAs can give a hint on many drivers and their interactions but axes loading does not go so as far as modern, explanatory machine learning can go. For example, while machine learning only looks at one driver at time, based on game theory, we can disentangle many drives. Moreover, since SHAP values for different drivers have the same unit as the response variable (here, flux units), we can still examine the 'relationship' between one driver and the response variable, namely using SHAP dependence plots (Fig. R2-1(a-c) for top three drivers for GPP). For example, with more light (PPFD) or higher soil temperature (TS), SHAP values of these drivers on GPP were generally increasing, suggesting a positive (for soil temperature then saturating) relationship between drivers and the response variables.

We will add partial dependence plots (Fig. R2-1) in the appendix as new Figure A3. Moreover, we will also add a description of this new figure in Section 3.3.

[Figure]

**Figure R2-1**. New Figure A3: SHAP dependence plots for top three drivers of GPP (upper row) and Reco (lower row)

In addition, we will also modify the following sentences in the method section:
(1) At current line 182 old sentence: "… (… Molnar, 2023). If a feature (i.e., a driver variable) has a positive SHAP value, this feature increases the local prediction relative to the overall mean prediction, and vice versa. Higher absolute…" will be modified to: "… (…Molnar, 2023). The SHAP value of a feature (i.e., a driver variable) represents the effect of that driver, at its specific value, on each individual prediction. A positive SHAP value indicates that this driver increases the local prediction to the overall mean prediction, and vice versa for negative SHAP values. Higher absolute …"
(2) At current line 187 old sentence: "… (… Qiu et al., 2022). In this study, we performed two SHAP analyses (1 and 2) with different foci." will be changed to: "… (… Qiu et al., 2022). The relationship between each driver and its SHAP values can be further explored using SHAP dependence plots. In this study, we performed two SHAP analyses (1 and 2) with different foci."

G4: Although the text introduces it correctly, I first falsely assumed that SHAP analysis 1 was based on RP, rather than on days, while SHAP 2 was on a daily basis. Just to confirm, is it correct that in both cases the SHAP analysis was performed on a daily basis but the daily results from SHAP 1 were then presented as RP averages in Fig. 4? For the next points, I presume that this is correctly understood.

Yes, both SHAP analyses were done on a daily basis. In fact, both SHAP analyses for GPP were based on the same XGBoost model, which was based on daily data. For SHAP analysis 1, we indeed averaged all SHAP values based on regrowth period, and for SHAP analysis 2, we showed daily values with a focus on extreme summer months. We will clarify this in the method section and make it clearer in the respective figure captions. At current line 198, we will add

"… extreme events. Both SHAP analysis 1 and SHAP analysis 2 were based on the same XGBoost model for GPP, which was performed based on daily data."

G5: I was surprised by the apparent lack of coherence between SHAP 1 and SHAP 2 analysis. Doesn't this show how sensitive the results are to the choice of the baseline (and of course the variables)? This should be mentioned when interpreting the results from such analysis.

Thank you for the comment. As mentioned above, these two analyses were based on the same set of variables and one XGBoost model. You are correct that any SHAP analysis strongly depends on the baseline used. For our two SHAP analyses 1 and 2, different baseline datasets were used, to provide insight into two different research questions, namely what are the main drivers under different environmental conditions, as explicitly mentioned in our objective 3.

Here, we provide an analogy: When you want to determine the factors that influence people's height, the results will naturally be different if you compare small infants with all male adults, or if you compare infants with tall basketball players. Similarly, the change of results in our analyses is also rather intuitive to understand. If you compare drivers of only summer GPP vs. drivers of two decades of GPP data (including winters), the variation in daily GPP is more dependent on the light/temperature difference. However, if you compare extreme summer GPP to non-extreme summer GPP with similar light/temperature conditions, you can actually see the effect of other variables (e.g., SWC, VPD) more clearly. So, yes, we fully agree that this nicely shows the sensitivity of such analyses to the chosen baseline. Therefore, we already in the earlier version of the manuscript described the baseline in Section 2.4.2. However, unfortunately, studies using machine learning models and SHAP explainers to understand fluxes have rarely mentioned these baseline datasets.

In order to raise attention to this aspect, we will add two new sentences in the beginning of Section 4.3 (line 430): "Using non-extreme summer months as the background dataset, SHAP analysis 2 focused on drivers of GPP only during extreme summer months, which minimized the confounding effects of strong seasonality in temperature and light conditions. This further improves the interpretability of the results, by isolating and focusing on the influence of extreme-related factors such as SWC and VPD. Across all extreme periods, …"

G6: Possible a priori relationship between response and driver variables: Defining RPs that can be compared across seasons and years seems a very appropriate approach, however, the RPs are of different lengths (see e.g. Fig. 6, where one RP spans over three months in 2018, while others span only over one month in 2022 and 2023). I wonder whether common relationships between driver variables and the RP length cause some artificial interdependence (circular logic) among some driver variables and, and even more worrisome, among some drivers and the response variable GPP.

I mean especially the two drivers DaySinceUse and DailyN, which represent management that both depend RP length. From a farmers perspective, i.e. planning the RP length to reach a certain goal, the RP length is inversely related to productivity and thus response variable daily GPP level.

The particular relationship between DaySinceUse and RP length is that only in cases when RP length is high, DaySinceUse can reach high values. In these cases, high DaySinceUse values coincide with low daily GPP and high RP length.

The particular relationship between DailyN and RP length is a mathematical consequence of the definition of DailyN , i.e. the DailyN value per unit fertilized N will inversely decrease with increasing RP length.

Please comment on the possible effects on the results of the analyses from these interdependencies or consider amelioration by different driver definitions.

Thank you for the constructive comments. The main goal of our XGBoost models was to accurately model GPP and Reco using the available main drivers. Since this grassland is intensively managed, DaySinceUse for mowing/grazing and DailyN for fertilization are two key variables representing management. To avoid any misunderstanding: the length of regrowth periods was never included in the model. In addition, the length of regrowth periods is not necessarily inversely relative to productivity, we rather deal with a cumulative effect. As we can see in the partial dependence plot (Fig. R2-1(c) for GPP), SHAP values for DaySinceUse are normally very negative at the beginning of regrowth periods (DaySinceUse < around 10 days) and then increase. Therefore, higher DaySinceUse values do not mean low daily GPP, but rather the opposite, high daily GPP values. As mentioned above, we will add this partial dependence plot for DaySinceUse together with other drivers in the appendix as new Fig. A3.

For more details about the calculation of DailyN, please see our response below (G9).

Regarding the length of regrowth period (also for comment D12): we have shown the relationship between the length of regrowth period and GPP/Reco below (Fig. R2-2). Overall, with longer regrowth periods, GPP or Reco during the regrowth period gets lower, which is to be expected when comparing spring-summer seasons versus autumn-winter seasons. Within different seasons, longer regrowth period does not always correspond to lower GPP or Reco. We will add more details on this in Section 3.2.

[Figure]

**Figure R2-2**. Length of regrowth periods vs GPP (left) and Reco (right).

Taken together with comment D19, we will also add statistics about the length at current line 251 as "The average length of the regrowth periods was 58 (± 47) days, with shorter regrowth periods in spring-summer seasons (39 ± 20 days) and longer in autumn-winter seasons (100 ± 60 days). A slight increasing trend (p = 0.01) was found in the length of regrowth periods over the 20 years. During spring-summer seasons, the length significantly increased (p < 0.01), while no significant trend was found in autumn-winter seasons (p = 0.87)."

G7: Critical reflection of using time as a driver: The variable DaySinceUse increases linearly with time until reaching RP length, i.e. it simply represents time as such or canopy age. The analysed relationship is thus the timeline for the development of GPP during an RP.

I wonder, what is the rationale behind using time as a driver? Short development time coincides with high productivity, but the drivers for production are not time, but rather the growth conditions. It will be obvious that the XGB-SHAP analysis will 'turn' time into a driver owing to the a priori decision of the user defining time as a driver.
Please explain the usefulness of time as a driver.

The referee's comment is based on a strong assumption: short regrowth period = high GPP, since a short regrowth period means a low number for the driver DaySinceUse. However, this assumption can be rejected since GPP during the regrowth period does not only relate to the length of the regrowth period, but does also relate to other meteorological and management variables, as seen from our model output (Figure A3). As shown in the partial dependence plot above, short development time (low DaySinceUse) does not correspond to high productivity. The length of the regrowth period is dynamic and can also change with other drivers, for example, farmers could adjust the mowing time during extreme events to "rescue" a regrowth before it dries out during a heatwave or before it gets soaked during a prolonged rainy period.

Nevertheless, it is indeed important in what form one inputs the drivers into the machine learning model and there are different ways of doing that (e.g. as continuous, categorical, binary variable, etc.). The aim behind the "days since" approach is to actually represent the management in a way the model understands. It is **not** exactly equal to time, which would be always increasing, while the DaySinceUse variable goes to zero each time there is a management event. Using "days since" as a driver actually gave reliable results in previous studies (Maier et al., 2022: time since management; Feigenwinter et al., 2023b: days since defoliation), but has also been used in studies by other groups, who used different machine learning models for gapfilling $N_2O$ fluxes (Goodrich et al., 2021: days since grazing).

G8: The above problem raises some fundamental questions about the general meaning of driver and response variables. The term adaptive management suggests that management, e.g. defining the time for harvest, i.e. the RP length, can both be a driver and response – a clear distinction may even be impossible.
What do these uncertainties mean for the interpretation of the analysis?
Is using the term "driver" together with a statistical analysis that is not able to detect cause-effect relationships (just effects) at all appropriate?

Thank you for the comment. As stated above, in our XGBoost models, length of regrowth periods was never included as a driver. The term 'adaptive' clearly indicates that management can both be a driver and a response, which is similar like the chicken and egg problem. Indeed a clear distinction is not possible. However, with no significant trends in GPP during regrowth periods observed, our evidence suggested that adapted management at this site has been able to maintain productivity even with on-going climate change as seen in more extreme events in recent years.

Another example of an environmental factor being both, driver and response, is light. Nobody would argue with the notion that light is driving plant photosynthesis (interacting with many other drivers, from water to nutrients), as is management. With more light (and days since management), photosynthesis increases, the plants grow, and the stand becomes taller and denser. As a consequence, or call it a response, stand structure changes, light attenuation

increases and light within the stand is decreasing, limiting photosynthesis compared to before. Thus, light is a driver and a response, as is management.

With machine learning approaches (here XGBoost + SHAP), we can identify top drivers of fluxes and assess their temporal development. These drivers were selected based on their known functional relevance for ecosystem processes. Even with traditional statistical analysis (e.g. linear relationships), a definite causal relationship cannot be detected. Therefore, we argue that the term 'driver' is appropriate in the manuscript.

G9: General reflection of the usefulness of DailyN as a driver for daily GPP: If I am right, DailyN is the only driver variable that includes averaging over the RP length. For a SHAP analysis that is based on daily values, the definition of DailyN is counterintuitive and the naming does not reflect what is actually going on (the fertilization is not daily). The authors will agree that N-availability might be the more relevant factor for GPP. N-availability will be larger right after the fertilization event (after reaching the rooting zone) and will decrease maybe not with time but with growth (and leaching, emissions etc.) over RP. Do you see a possibility to define an alternative daily variable "available N" (AN) that parameterizes the decrease from the amount of fertilized N over the length of the RP or even scaling it negatively with GPP (as proxy for growth)?

Thank you for the comment. How to best represent N fertilization and "available N" in any model is indeed an ongoing discussion, also in our group. We agree that the current value can be discussed, but currently we consider this as our best approach. During the majority of regrowth periods, there was only one fertilization event, and normally it happened relatively soon after the mowing. If there was a second fertilization event, we summed up the amount of fertilized nitrogen for that period and calculated DailyN based on the total amount. This way, we consider the supplied N as fully available for growth. To be extremely accurate, one would need to model the available N as suggested, e.g., with an exponential decline, considering many factors like soil properties, precipitation events and soil moisture dynamics, but also microbial activity, potentially delayed responses by soil moisture and soil temperature. Achieving an accurate estimate of this on a daily scale would either need another exhaustive study (with unknown methodology if data were to be collected daily in a real-world setting, not in a lysimeter) or would be based on many untested assumptions. Thus, we cannot solve this issue by changing the regrowth period average to an exponential decay or other dynamic functions without introducing large untested (maybe even untestable) bias. Furthermore, we did not find convincing alternative solutions based on the existing studies in the literature, which are currently still limited. Therefore, we consider our current way of representing real management as best as we can do, but would of course welcome more systematic approaches in future work.

G10: In general, please explain the term adaptive climate smart decisions / management. This is important for two reasons, i) it is used in the interpretation and the conclusions and ii) the definition might help to better understand the nature of the RP, i.e. as depending on certain a priory rules and expectations/ observations on productivity.

Thank you for the suggestion. As also suggested by Referee 1, We will introduce more explanation on this term in current line 69.

G11: The study concludes (L474-L476) adaptive climate smart management as a factor for homogeneous production despite weather trends (likely climate change induced). Is this 'just' a plausible speculation or did your study show this? Maybe I overlooked it, I did not find clear evidence in the presented results for this statement that comes up in the discussion, the

conclusions and is highlighted in the abstract. A quantitative analysis would examine the interaction between adaptive climate smart management and production, probably in contrast to a plausible BAU scenario.

I wonder whether XGB generated predictions could be used for scenario calculations or whether mechanistic models would be needed to substantiate such speculation. I deem this worth to be clarified in the discussion. I do not suggest such study to be included here. The study is rich enough, but its limitations need careful consideration, i.e. what can be concluded from its results.

Thank you for the constructive comment. With more frequent extreme events in recent years that were observed in our time series, we expected a decreasing trend in GPP during regrowth periods and ultimately lower $CO_2$ sink strength, unless management was already adapted to these new conditions. The non-significant trend detected in our GPP data clearly showed that the existing management practices were able to maintain productivity, thus suggesting resilience to extreme events through 'climate-smart' management, following the definitions used by science as well as global organizations such as IPCC, FAO and the World Bank. We agree that more "climate-smart" management practices aiming to improve resilience and sustainability in agroecosystems should be tested in the field with experiments or with well calibrated process models (beyond machine learning models), e.g., testing the effect of timing and intensity of certain management practices on productivity and GHG emissions. We have ongoing work in the group using the process-based model MONICA (Nendel et al., 2011) on this exact topic (Kamali et al., submitted).

With all considerations in mind, as also suggested by Referee 1, we will put our argumentation in context and explain this aspect better throughout the manuscript. For the conclusion section, we will modify lines 474-476 as "Moreover, based on two decades of measurements, our evidence suggests that the grassland farmer succeeded in managing the site with relatively stable GPP during regrowth periods, based on climate-smart adaptive management."

General recommendation: I deem the overall quality of the manuscript to be very high and inspiring and maybe its clarity is the reason why it provokes some critical thoughts. I do not claim that this review from reading the manuscript a couple of times, can be assumed to be exhaustive and accurate, as I lack particular knowledge that the Authors probably have. I expect though clarifying responses and look forward to the answers by the authors. It depends very much on these answers, whether minor or major revisions will be necessary.

Detailed comments (please comment and take action, where applicable)

D1: The title includes the word regrowth, which implies biomass production while it is used here as re-establishment or recovery of GPP and Reco. I suggest using the more neutral "grassland $CO_2$ exchange:" instead of "grassland $CO_2$ fluxes and regrowth:"

Thank you for the comment. We will change the title into "Drivers of long-term grassland $CO_2$ fluxes: effects of management and meteorological conditions during regrowth periods".

D2: L 16: consider starting a new paragraph before "$CO_2$".

We can do this if the journal allows two separate paragraphs in the abstract. We can modify the sentence as "Our results showed pronounced … in $CO_2$ fluxes, driven by both …".

D3: L 20 and L 25: make a decision on whether the study showed or suggested a relationship between $CO_2$ exchange and "adapted, climate smart decision making" (see also G11).

We will make sure we stay consistent with 'suggested'.

D4: L53-54: Define "atmospheric dryness" – explain, why does it not include "reduced precipitation".

We define "atmospheric dryness" as high VPD, which could be independent from reduced precipitation. We will add this definition in brackets.

D5: L56-L57: While promote productivity makes sense "promote $CO_2$ fluxes in general" does not– consider rewording.

The studies cited found both increased GPP and Reco. We will modify the sentence to "… that warming can promote grassland productivity and increase $CO_2$ fluxes (both GPP and Reco) …"

D6: L64: is the word 'buffer' appropriate here? 'mitigate impact of … on …'?

'Buffer' was used in the original study. We will change the sentence into "… may mitigate the impact of temperature anomalies on $CO_2$ fluxes" to improve clarity.

D7: L73 – L77: Please clarify, do you mean the nonlinear, interactive, and highly dynamic "nature of drivers" or rather the nonlinear, interactive, and highly dynamic "nature of responses"? Please consider the difference between "dynamic" and, e.g. "variable"? What would fit better here?

Here, we meant for both drivers themselves and their effects. We will modify the sentence as "Linear models frequently fail to capture the nonlinear, interactive, and highly variable effects of drivers that influence $CO_2$…"

D8: L84: (objective 1) Do you deem the investigation of something as a scientific objective?

We will use the word "identify".

D9: L96 – what do you mean with "destroyed"? was just ploughed, or extracted and removed?

The sward was killed by either direct ploughing everything under (2012) or herbicide application (2021). We will modify as "the existing sward was terminated and ploughed…"

D10: L103-L107: If I am right, this is an important decision on how to use and interpret ¼ of the time series. I wonder how this decision has influenced the results. I would like you to discuss the alternative(s), e.g. separating fluxes between parcels and using only the comparable one, parcel B, for this study.

In general, the management regimes (in terms of mowing and grazing dates) were very similar between these two parcels, also during the $N_2O$ mitigation experiment. Differences in management happened in earlier years (2008-2010), and most of the cases were that one parcel was grazed while one was mown, but both around the same time (see management info in Table B2 in Feigenwinter et al., 2023a). In addition, the parcel areas changed twice in the past 20 years, and in earlier years parcel B dominated the footprint area (Figure B3 in Feigenwinter et

al., 2023a). Since different wind directions dominate during days and nights, separating fluxes will significantly decrease our data coverage and create bias since we use daily data in the analysis. In our previous study at the same site regarding long-term carbon budgets (Feigenwinter et al., 2023a), we did not separate the parcels either. We have taken the different fertilization regimes during the $N_2O$ mitigation experiment into account when calculating DailyN (lines 115-116) so that the whole field received overall less N fertilization. For all these reasons, we will continue using regrowth periods based on parcel B.

D11: L109-112: Please specify "This" in "This allows" – the logic between the two sentences is not clear (to me). Did you merge shorter periods into one RP? If this was the case, did you check, whether the results in these RP differed from the others?

In earlier years (2008-2010), some parts of the field were first grazed, and some days (e.g., five days) later the other parts were mown. If we account these short periods as regrowth periods, we would have five more regrowth periods than the current 115 regrowth periods. Since all our analyses are based on the entire field, separating the field is not possible (see our answer to D10). Even if we would define these five days between grazing and mowing events as one regrowth period although it only occurred at parts of the field, we could not reliably calculate GPP of the entire field, since we capture the combined signal from the entire footprint area (i.e. areas with short vegetation and areas with high vegetation). Thus, to make this clearer, we will modify the sentence as "If management activities took place only a few days apart within the entire footprint, we defined the later activity as the end of the regrowth period."

D12: L112: Be aware of the impact that averaging over differently long RP has on the meaning of the variable. Is there a negative relationship between the length of the RP and the average GPP or Reco value? Or has the definition of a minimum RP length of 10 days alleviated this relationship. From my distant perspective, if such relationship existed, it would explain relationships between effects from drivers that are (Day[s]SinceUse, dailyN) or are not (PAR, TA, TS, VPD) related to the length of the RP (see also G6).

Thank you for the comment. Please see our response to G6 above.

D13: L 109: Please add a short explanation on how was the position and length of the main "growing season" defined, and how this RP classification affected the analysis. Consider replacing 'middle date' by 'center'

This classification is based on local management, also typical for other central European grasslands, as seen in Figure A2 (which will become Fig. 3a). Usually the first mowing event of the year happens in April and the last mowing happens in mid/end October Thus, this is the "main growing season", mainly used for visualization purposes in Fig. 4, and to test the trend in GPP and Reco in different seasons. We will modify the sentence at line 119 as "If the center of one regrowth period was between April and September (which is the main growing season for agricultural grasslands in central Europe), the period was classified as …". We will also change "middle date" to "center" and change the figure caption accordingly.

D14: L125: consider "atmospheric" instead of "air"

We will change this accordingly.

D15: L128: specify, probably, 'volumetric' SWC

Thank you for the reminder. We will change to "volumetric SWC".

D16: L130: replace "community guidelines" with scientific references

We will specify this sentence as "Widely used community guidelines (Aubinet et al., 2012; Pastorello et al., 2020, Sabbatini et al., 2018), including …".

D17: L138: explain why using these percentiles instead of the usual ones 5 %, 50 % , and 95 %?

The $16^{th}$ and $84^{th}$ percentiles are statistically equivalent to $\pm 1$ standard deviation ($\sigma$) for data that follow a normal distribution. This approach is consistent with the standardized FLUXNET processing pipeline (Pastorello et al., 2020), where these percentiles are used to calculate overall uncertainty.

D18: L147-L149: Please explain this averaging choice considering the vertical distributions of roots and SOM.

Thank you for the comment. We do not know the exact rooting depths of this grassland nor the exact depths of plant water uptake. Rooting and water uptake depths could also differ due to heterogeneity within the field. Furthermore, it is uncertain which soil depths are responsible for ecosystem respiration. With the EC measurements, flux data represent the integrated condition over the entire field. Therefore, we averaged the soil variables across depths to represent the integrated soil conditions over the entire soil profile. We will adjust this sentence as "For soil temperature… averages across all depths were calculate and used in the final analysis to represent the overall soil conditions over the entire profile."

D19: L154 – L155: If only the GPP Reco averages have been tested, please add information, on whether the lengths of the RP showed a trend. Please specify: in the trend analysis did you exclude extreme months?

We did not exclude extreme months in our trend analysis. We have tested the length of the regrowth periods. Overall, the length of regrowth periods had a slight increasing trend (p = 0.01). During spring-summer seasons, the length significantly increased (p < 0.01), while no significant trend was found in autumn-winter seasons (p = 0.87). We will include this information in Section 3.2. Please also see our response to G6.

D20: L165: Please clarify, what do you mean with "GPP regrowth rate" do you mean dGPP/dt or, in accordance what was explained above, "average GPP rates over RPs"? Then please explain the rationale for choice of a 2nd order polynomial as regression model for the analysis of light response function of GPP.

We meant average GPP during the regrowth periods. For each regrowth period, cumulative GPP and Reco were first calculated and then averaged based on the length of the regrowth period. Following the suggestions from Referee 1, we will adjust the term of 'GPP/Reco regrowth rate' to "GPP/Reco during regrowth periods" throughout the text. We chose a second order polynomial to model the saturating relationship between light and GPP.

D21: L168-L172: Specify when adding GPP as driver variable for Reco, why did you still include PPFD and VPD as drivers? Is it correct to say that that all other effects are then residual effects, i.e. effects of a variable on top of its effect on GPP?

We added GPP as a driver for Reco model, since GPP provides the carbohydrates for carbon allocation belowground, for root systems as well as microbes, thus for auto – and heterotrophic respiration. We kept the other drivers in the Reco driver analysis to keep the drivers of the two models (GPP, Reco) as similar as possible.

No, it is not correct to say that these drivers have residual effects after accounting for GPP effects. In the XGBoost model and SHAP analyses, the effects of certain drivers are not comparable to, for example, linear regressions. Instead, the effects are additive based on the nature of SHAP values (Lundberg et al., 2020), see our answer above to G2. Keeping PPFD and VPD in the model will not take away any statistical power of other drivers. We will modify the sentence as "… GPP was added as an additional feature to better represent the carbon supply and allocation for autotrophic and heterotrophic respiration.".

D22: L197 – The sentence does not make sense in the way that in both set-ups the same months (JJA) were selected without further distinction. Is in the sentence describing the second set after "for only the peak growing season" a reference to extreme years missing? Or did I misunderstand anything here?

Sorry for the confusion, and indeed there was some information missing. We will clarify the sentence as "Here, we used only the months during the peak growing season (June, July, August) as the background dataset when no extreme weather condition occurred. We then calculated a second set of SHAP values for only the peak growing season in 2018, 2019, 2022, 2023, as these periods included the majority of recent extreme months.". Regarding the background dataset, please see our answer above to G4.

D23: L181 - L184: Clarify here that an effect is different from a relationship (see G2)

See our response to G2 and G3.

D24: Section 3.1: Would the length of the vegetation period (VP) and the meteorological conditions in the VP - both more relevant for bioclimatological characterization- give a different picture? Consider moving Table 1 in the appendix and focusing only on significant trends here.

Thank you for the comment. In permanent grasslands, the vegetation period, as seen in the regrowth periods, can be year around. Since regrowth periods can differ from year to year and are not aligned to calendar months, we provide the bioclimatological characterization on a monthly basis, which also allows comparison to other sites in Europe. We will move Table 1 to the appendix as the new Table A2.

D25: Figure 2: Nice and clear presentation and reasoning.

Thank you for the positive comment.

D26: Section 3.2: can you provide information about interannual and seasonal variation of the length of RP (e.g. horizontal range bars or replace circle by rounded rectangles in Figure 3). Alternatively you might consider presenting the sums of GPP and Reco over RPs as alternative to the average in the same manner in a second figure. It might show, if I am right, the effects of adaptive management.

We will add statistics on the length of regrowth period in the main text. Please also see our answers above to G6 and D12. Since the new Figure 3 (Fig. R2-3) will be expanded following

suggestions from Referee 1, we will not add further information in this figure. The magnitude of GPP and Reco is already depicted in Figure 3, as the fluxes are given by the color code.

[Figure]

**Figure R2-3**. New Figure 3 for the revised manuscript

D27: Section 3.3: Fig. 4: I find the choice of the transparent colors confusing because they do not match with the colors of the legend very well. Consider an alternative to show the different season categories if at all necessary. Mark in the figure when the grassland renewal has taken place and which were the months with extreme weather. Then explain that the x-axis variable is not time but number of RP from start of the investigated period.

Thanks for your suggestion, but we thought a lot about the color scheme and how best to convey such complex information as in Figure 4. Therefore, we use the same color for the same driver but more transparent for the fall/winter season (which is shorter than the spring/summer season). For this SHAP analysis 1, we used two decades of data and present the overall picture, thus seasonality is important to depict. We already stated in the figure caption that each stacked bar represents different regrowth periods. To allow comparison with other figures for the same year, we cannot change the x-axis. The number of regrowth periods within each year can be clearly seen. In addition, stats on the regrowth periods will be given in a revised manuscript (see answer

to D26). Since this figure is already busy, we cannot put more info in there, such as renewals or extreme weather. For the latter, we provide Figures. 5-6. We will try to improve the legend for different seasons in Fig.4.

D28: Section 3.4: Make sure to mention that the difference between the canopy photosynthesis saturation level and the maximum of a polynomial GPP= f(PAR) have different meanings (see also D20).

We did already specify in line 298 "(GPPmax, i.e. the maximum of the curves in Figure 5)". We now avoid the term saturation to make it more clear and changed the sentence to: "During all periods, daily GPP increased with mean daily PPFD, while maximum GPP (GPPmax, i.e., the maximum of the curves in Figure 5) was reached at a PPFD of about 30 mol m$^{-2}$ day$^{-1}$ (before the renewal years) or at about 45 mol m$^{-2}$ day$^{-1}$ (after the renewal years, all other normal years).

D29: L277-279: 'negative effects of SWC on Reco' is a very good example, how effects may sound counterintuitive. I think it would be good mentioning, recalling that SWC is low during droughts, low values of SWC have caused the low predicted Reco as indicated by the negative SHAP effect values (see also G2).

See our response to G2 and G3. We will modify this sentence into "During extreme summers when low SWC occurred, negative effects of SWC on Reco were more obvious in 2022 and 2023 compared to earlier years…"

D30: Section 3.5: the heading focuses on drivers, but, I believe, the main relevant results are the effects on GPP and Reco.

Second 3.5 is about drivers of GPP in extreme summers, not about Reco.

**References**
Aubinet, M., Vesala, T., and Papale, D. (Eds.): Eddy Covariance: A Practical Guide to Measurement and Data Analysis, Springer Netherlands, https://doi.org/10.1007/978-94-007-2351-1, 2012.

Feigenwinter, I., Hörtnagl, L., Zeeman, M. J., Eugster, W., Fuchs, K., Merbold, L., and Buchmann, N.: Large inter-annual variation in carbon sink strength of a permanent grassland over 16 years: Impacts of management practices and climate, Agric. For. Meteorol., 340, 109613, https://doi.org/10.1016/j.agrformet.2023.109613, 2023a.

Feigenwinter, I., Hörtnagl, L., and Buchmann, N.: $N_2O$ and $CH_4$ fluxes from intensively managed grassland: The importance of biological and environmental drivers vs. management, Sci. Total Environ., 903, 166389, https://doi.org/10.1016/j.scitotenv.2023.166389, 2023b.

Goodrich, J. P., Wall, A. M., Campbell, D. I., Fletcher, D., Wecking, A. R., and Schipper, L. A.: Improved gap filling approach and uncertainty estimation for eddy covariance $N_2O$ fluxes, Agric. For. Meteorol., 297, 108280, https://doi.org/10.1016/j.agrformet.2020.108280, 2021.

Kamali, B., Buchmann, N., Feigenwinter, I., Wang, Y., Ewert, F., Gaiser, T.: Navigating the trade-off among biomass production and GHG emissions for smart management of grasslands, Submitted.

Krebs, L., Hörtnagl, L., Scapucci, L., Gharun, M., Feigenwinter, I., and Buchmann, N.: Net ecosystem $CO_2$ exchange of a subalpine spruce forest in Switzerland over 26 Years:

Effects of phenology and contributions of abiotic drivers at daily time scales, Global Change Biol., 31, e70371, https://doi.org/10.1111/gcb.70371, 2025.

Li, X., Ciais, P., Fensholt, R., Chave, J., Sitch, S., Canadell, J. G., Brandt, M., Fan, L., Xiao, X., Tao, S., Wang, H., Albergel, C., Yang, H., Frappart, F., Wang, M., Bastos, A., Maisongrande, P., Qin, Y., Xing, Z., Cui, T., Yu, L., He, L., Zheng, Y., Liu, X., Liu, Y., De Truchis, A., and Wigneron, J.-P.: Large live biomass carbon losses from droughts in the northern temperate ecosystems during 2016-2022, Nat. Commun., 16, 4980, https://doi.org/10.1038/s41467-025-59999-2, 2025.

Liu, J., Wang, Q., Zhan, W., Lian, X., and Gentine, P.: When and where soil dryness matters to ecosystem photosynthesis, Nat. Plants, 11, 1390–1400, https://doi.org/10.1038/s41477-025-02024-7, 2025.

Lundberg, S. M., Erion, G., Chen, H., DeGrave, A., Prutkin, J. M., Nair, B., Katz, R., Himmelfarb, J., Bansal, N., and Lee, S.-I.: From local explanations to global understanding with explainable AI for trees, Nat. Mach. Intell., 2, 56–67, https://doi.org/10.1038/s42256-019-0138-9, 2020.

Maier, R., Hörtnagl, L., and Buchmann, N.: Greenhouse gas fluxes ($CO_2$, $N_2O$ and $CH_4$) of pea and maize during two cropping seasons: Drivers, budgets, and emission factors for nitrous oxide, Sci. Total Environ., 849, 157541, https://doi.org/10.1016/j.scitotenv.2022.157541, 2022.

Nendel, C., Berg, M., Kersebaum, K. C., Mirschel, W., Specka, X., Wegehenkel, M., Wenkel, K. O., and Wieland, R.: The MONICA model: Testing predictability for crop growth, soil moisture and nitrogen dynamics, Ecol. Modell., 222, 1614–1625, https://doi.org/10.1016/j.ecolmodel.2011.02.018, 2011.

Novick, K. A., Ficklin, D. L., Grossiord, C., Konings, A. G., Martínez‐Vilalta, J., Sadok, W., Trugman, A. T., Williams, A. P., Wright, A. J., Abatzoglou, J. T., Dannenberg, M. P., Gentine, P., Guan, K., Johnston, M. R., Lowman, L. E. L., Moore, D. J. P., and McDowell, N. G.: The impacts of rising vapour pressure deficit in natural and managed ecosystems, Plant Cell Environ., 47, 3561–3589, https://doi.org/10.1111/pce.14846, 2024.

Pastorello, G., Trotta, C., Canfora, E., Chu, H., Christianson, D., Cheah, Y.-W., Poindexter, C., Chen, J., Elbashandy, A., Humphrey, M., Isaac, P., Polidori, D., Reichstein, M., Ribeca, A., van Ingen, C., Vuichard, N., Zhang, L., Amiro, B., Ammann, C., Arain, M. A., Ardö, J., Arkebauer, T., Arndt, S. K., Arriga, N., Aubinet, M., Aurela, M., Baldocchi, D., Barr, A., Beamesderfer, E., Marchesini, L. B., Bergeron, O., Beringer, J., Bernhofer, C., Berveiller, D., Billesbach, D., Black, T. A., Blanken, P. D., Bohrer, G., Boike, J., Bolstad, P. V., Bonal, D., Bonnefond, J.-M., Bowling, D. R., Bracho, R., Brodeur, J., Brümmer, C., Buchmann, N., Burban, B., Burns, S. P., Buysse, P., Cale, P., Cavagna, M., Cellier, P., Chen, S., Chini, I., Christensen, T. R., Cleverly, J., Collalti, A., Consalvo, C., Cook, B. D., Cook, D., Coursolle, C., Cremonese, E., Curtis, P. S., D'Andrea, E., da Rocha, H., Dai, X., Davis, K. J., Cinti, B. D., Grandcourt, A. de, Ligne, A. D., De Oliveira, R. C., Delpierre, N., Desai, A. R., Di Bella, C. M., Tommasi, P. di, Dolman, H., Domingo, F., Dong, G., Dore, S., Duce, P., Dufrêne, E., Dunn, A., Dušek, J., Eamus, D., Eichelmann, U., ElKhidir, H. A. M., Eugster, W., Ewenz, C. M., Ewers, B., Famulari, D., Fares, S., Feigenwinter, I., Feitz, A., Fensholt, R., Filippa, G., Fischer, M., Frank, J., Galvagno, M., et al.: The FLUXNET2015 dataset and the ONEFlux processing pipeline for eddy covariance data, Sci. Data, 7, 225, https://doi.org/10.1038/s41597-020-0534-3, 2020.

Sabbatini, S., Mammarella, I., Arriga, N., Fratini, G., Graf, A., Hörtnagl, L., Ibrom, A., Longdoz, B., Mauder, M., Merbold, L., Metzger, S., Montagnani, L., Pitacco, A., Rebmann, C., Sedlák, P., Šigut, L., Vitale, D., and Papale, D.: Eddy covariance raw data processing

for $CO_2$ and energy fluxes calculation at ICOS ecosystem stations, Int. Agrophys., 32, 495–515, https://doi.org/10.1515/intag-2017-0043, 2018.

Shekhar, A., Hörtnagl, L., Buchmann, N., and Gharun, M.: Long-term changes in forest response to extreme atmospheric dryness, Global Change Biol., 29, 5379–5396, https://doi.org/10.1111/gcb.16846, 2023.

---

## Author Comment (AC3)

**Author Comments – Response to Referee 3**

Referee comments are marked in black and author responses are marked in blue.

Grasslands constitute a significant portion of agricultural landscapes in Europe and store substantial amounts of carbon. Effective management of these ecosystems serves as a nature-based solution for enhancing carbon sequestration. This study presents a unique long-term dataset spanning 20 years, encompassing flux, biometeorological, and detailed management data from an eddy covariance site in a Swiss grassland. It addresses the gap in understanding the temporal dynamics and development of $CO_2$ drivers by employing machine learning models, as opposed to the predominantly linear models used previously.

This research contributes to a detailed understanding of the impact of various management practices on the carbon budgets of mid-latitude grasslands. The use of state-of-the-art machine learning models provides an extended understanding of non-linear and dynamic relationships between $CO_2$ flux and its drivers. The study is methodologically sound, with comprehensive descriptions of the eddy covariance setup, the data processing, and statistical analyses. However, some concepts require further clarification. The results are well discussed, figures are well-described and visually comprehensible.

Dear Referee,
Thank you for your positive feedback! We appreciate the opportunity to improve the manuscript based on your thorough review and constructive comments. We have addressed your questions and suggestions in the responses below.

Specific comments:

Methods & Results:
1. Line 147: the soil variables are averaged across depths - what is the reasoning behind that? Especially for the soil water content measurements were taken in very different depths, and soil water fulfill different functions in the top soil layer than in the deep soil layer, depending on how available it is to microbes or plant roots-> why not use separately and see which layer is important in which times to gain more process understanding?

Thank you for the comment. The exact rooting depths of this grassland are unknown, as well as the exact depths of plant water uptake. Rooting and water uptake depths could also differ due to heterogeneity within the field. Furthermore, it is uncertain which soil depths are responsible for ecosystem respiration. With the EC measurements, flux data represent the integrated condition over the entire field. Therefore, we averaged the soil variables across depths to represent the integrated soil conditions over the entire profile, instead of focusing on specific depths. Nevertheless, we will adjust this sentence as "For soil temperature… averages across all depths were calculate and used in the final analysis to represent the overall soil conditions over the entire profile."

2. Line 114-115: DailyN is calculated as a daily average across the regrowth period. However, not the same amount of N would be available to the plants each day, maybe in the beginning more and later less, depending also on environmental factors such as strong rain events and leaching – please elaborate in how far this has implications on your results.

Thank you for the comment. As we responded to Referee 2, how to best represent N fertilization and "available N" in any model is indeed an ongoing discussion, also in our group. We agree

that the current value can be discussed, but currently we consider this as our best approach. During the majority of regrowth periods, there was only one fertilization event, and normally it happened relatively soon after the mowing. If there was a second fertilization event, we summed up the amount of fertilized nitrogen for that period and calculated DailyN based on the total amount. This way, we consider the supplied N as fully available for growth. To be extremely accurate, one would need to model the available N as suggested, e.g., with an exponential decline, considering many factors like soil properties, precipitation events and soil moisture dynamics, but also microbial activity, potentially delayed responses by soil moisture and soil temperature. Achieving an accurate estimate of this on a daily scale would either need another exhaustive study (with unknown methodology if data were to be collected daily in a real-world setting, not in a lysimeter) or would be based on many untested assumptions. Thus, we cannot solve this issue by changing the regrowth period average to an exponential decay or other dynamic functions without introducing large untested (maybe even untestable) bias. Furthermore, we did not find convincing alternative solutions based on the existing studies in the literature, which are currently still limited. Therefore, we consider our current way of representing real management as best as we can do, but would of course welcome more systematic approaches in future work.

3. Line 140: The gaps in flux data were filled with a random forest model. Might these confound the driver analysis that is conducted later? E.g. if gap-filling is based on other environmental variables such as Tair or VPD, of course the driver analysis shows high dependence of ecosystem fluxes on these variables. Please clarify, and if there might be confounding effects elaborate on that in the discussion.

Indeed, the half-hourly NEE data were gap-filled using a random forest model that takes into account a large number of factors, including timestamp (e.g., year, month, date, hour, DOY), meteorological variables that are used in traditional gap-filling marginal distribution sampling (MDS) methods (i.e., SW_IN, TA, VPD), and management events within the footprint (Hörtnagl, 2025), thus many more than were relevant in the driver analyses. We then partitioned the gap-filled NEE dataset using the nighttime partitioning method into GPP and Reco at half-hourly resolution. For the subsequent XGBoost models and SHAP analysis, we used daily aggregated values (mean or sum, depending on the variable). While we acknowledge the fact that certain variables are used in both processes (i.e., gap-filling and XGBoost models), the confounding effect is expected to be very small due to the following reasons:

(1) Beyond the meteorological variables, the random forest gap-filling also relies on temporal and management information. The gap-filling can capture short-term variability of the NEE, while the driver analysis focuses on overall driver importance based on broader temporal scales (20 years) for GPP and Reco.
(2) The driver analysis (XGBoost + SHAP) conducted based on daily aggregated data, therefore reducing potential correlation and confounding effects that might occur at finer time scales.
(3) In the driver analysis, PPFD_IN were used instead of SW_IN, and additional meteorological variables were used (e.g., TS, SWC). The variables used in the gap-filling process were found to have relatively low importance (current Figure A3b-d).

In addition, we cannot avoid gap-filling NEE: If we would use only measured, good quality data for the daily averages, we would overestimate the net $CO_2$ uptake as there are less data during the night when respiration is the dominant signal. We will add this information in Section 4.2 in the revised version of the manuscript.

4. Figure 4:
For the partitioning of GPP and Reco the nighttime partitioning method by Reichstein et al (2005) was used. In this method, this means, Reco is modelled depending on temperature. GPP is calculated as the difference between measured NEE and modelled Reco. Of course there will be dependencies then of the component fluxes to temperature and GPP, and their contribution in explaining the fluxes might be partly conceptually introduced. Does it "even out" since you integrate over regrowth periods?

I would suggest to quickly test another partitioning method (e.g. hybrid partitioning method introduced by Nguyen et al (2025), code available on github) to confirm if the main drivers are the same or if the results are confounded the partitioning method.
Nguyen, N. B., Migliavacca, M., Bassiouni, M., Baldocchi, D. D., Gherardi, L. A., Green, J. K., Papale, D., Reichstein, M., Cohrs, K.-H., Cescatti, A., Nguyen, T. D., Nguyen, H. H., Nguyen, Q. M., and Keenan, T. F.: Widespread underestimation of rain-induced soil carbon emissions from global drylands, Nat. Geosci., https://doi.org/10.1038/s41561-025-01754-9, 2025.

Thank you for the comment. During the preliminary exploratory data analysis, we compared different gap-filling methods, including nighttime (Reichstein et al., 2005), daytime (Lasslop et al., 2010), and the recent improved-daytime (Keenan et al., 2019) methods. In our PI dataset used in this study, we also included daytime partitioned fluxes as part of the flux product (Hörtnagl et al., 2025). On daily scale, the partitioned GPP and Reco had similar variation and dynamic/pattern, so we assume the result of our subsequent driver analysis to be similar as well.

In nighttime partitioning methods, GPP is calculated based on the difference between NEE and Reco, therefore it is less dependent on the variables used in modelling Reco. Thus, we chose to use this partitioning method for our driver analyses, also commonly used in other studies. Moreover, air temperature (TA) was used to model Reco in the nighttime partitioning methods. However, the XGBoost model for Reco used soil temperature (TS) as temperature-related driver, not TA. In the XGBoost model for GPP, TS turned out to be a better driver than TA in terms of importance (current Figure A3b), which also showed that the dependency of this component flux is stronger on TS than on TA.

The recent study by Nguyen et al. (2025) provides new insights, especially on machine-learning based partitioning methods. Our manuscript was submitted before the publication of this new method, therefore we did not include this method in the exploratory data analysis. We understand that the proposed partitioning method has a focus on global dryland, which is not the same ecozone/climate zone as this temperate grassland, and thus different processes are relevant in such semi-arid systems.

Technical corrections:

Line 95: how many times per year/ in which times fertilizer is applied? Please specify. One fertilizer application per regrowth period?

Yes, there was usually one fertilizer application per regrowth period (Fig. A2). The number of fertilizer applications differed among years, normally depending on the number of mowing events. We will update the sentence as "… (mainly as slurry, normally applied once after each mowing event…".

Line 128: "For more detailed information on instrumentation, see …"

Thank you for the suggestion. We will modify the sentence accordingly.

Line 205: mean annual temperature and precipitation is a repetition from lines 91-92, can be removed here or just put in brackets, e.g. "mild temperate climate (9.9 °C, 1147 mm)"

Thank you for the suggestion. We will shorten the sentence into "… mild temperate climate (MAT 9.9 °C, MAP 1147 mm)."

Line 354-356: the two parts of the sentences are not necessarily connected, I would suggest to split it up into two sentences. First part ("Often, studies focused on natural ecosystems or 355 forests at regional and global scales (Anav et al., 2015; Cai and Prentice, 2020; Davi et al., 2006; Norby et al., 2010)") fits better in the Introduction.

Thanks for the suggestion, but the info of the first sentence is indeed relevant for the second. To improve understanding, we will change the first sentence as "Often, studies focused on natural unmanaged ecosystems or forests…" to show that we want to emphasize the different ecosystems that have been studied.

Line 380-383: please improve for more clear language: "Such observations, albeit rare, are impactful in terms of C dynamics, and underscore the necessity to include such these infrequent destructive management practices in long-term flux studies and as well as modelling frameworks."

Thank you for the feedback. We will incorporate this change as suggested.

Line 392-393: sentence is unnecessarily long and not so well understandable, please split it up into two shorter sentences

We will change this sentence into two shorter sentences.

Line 428: what do you mean with "explanatory outputs which enhanced interpretability"? It sounds a bit generic, please specify

Here we meant the explainable machine learning tool like SHAP to improve the interpretability of the tree-based models. We will modify the sentence as "… but also provided explanatory outputs such as driver importance on a daily basis, which enhanced interpretability of the machine learning models and further improved…"

Line 436: in line 431 you state that during extreme events "SWC strongly reduced GPP", but later you state that "more significant negative effects of droughts on grassland $CO_2$ fluxes compared to those of heatwaves". It seems a bit contradictory, maybe you can clarify this section.

We meant to compare the effect of SWC and VPD. We will clarify this sentence as "… more significant negative effects of soil droughts (i.e., low SWC) on grassland $CO_2$ fluxes compared to those of heatwaves (i.e., high VPD) …"

Line 457-460: complicated sentence structure, please simplify or split into two sentences

We will split it into two sentences.

Line 462: please mention briefly what precision-farming is (and please mind consistent terms – in L.477 precision farming is written without "-")

Thank you for the reminder. We will add "… and spatially adjusted management practices such as precision-farming …".

**References**

Hörtnagl, L.: diive v0.87.1, https://doi.org/10.5281/zenodo.15648669, 2025.

Hörtnagl, L., Feigenwinter, I., Wang, Y., Buchmann, N., Merbold, L., Zeeman, M., Fuchs, K., and Eugster, W.: Eddy covariance ecosystem fluxes, meteorological data and detailed management information for the intensively managed grassland site Chamau in Switzerland, collected between 2005 and 2024: CH-CHA FP2025.3 (2005-2024), https://doi.org/10.3929/ethz-b-000747025, 2025.

Keenan, T. F., Migliavacca, M., Papale, D., Baldocchi, D., Reichstein, M., Torn, M., and Wutzler, T.: Widespread inhibition of daytime ecosystem respiration, Nat. Ecol. Evol., 3, 407–415, https://doi.org/10.1038/s41559-019-0809-2, 2019.

Lasslop, G., Reichstein, M., Papale, D., Richardson, A. D., Arneth, A., Barr, A., Stoy, P., and Wohlfahrt, G.: Separation of net ecosystem exchange into assimilation and respiration using a light response curve approach: critical issues and global evaluation, Global Change Biol., 16, 187–208, https://doi.org/10.1111/j.1365-2486.2009.02041.x, 2010.

Nguyen, N. B., Migliavacca, M., Bassiouni, M., Baldocchi, D. D., Gherardi, L. A., Green, J. K., Papale, D., Reichstein, M., Cohrs, K.-H., Cescatti, A., Nguyen, T. D., Nguyen, H. H., Nguyen, Q. M., and Keenan, T. F.: Widespread underestimation of rain-induced soil carbon emissions from global drylands, Nat. Geosci., 18, 869–876, https://doi.org/10.1038/s41561-025-01754-9, 2025.

Reichstein, M., Falge, E., Baldocchi, D., Papale, D., Aubinet, M., Berbigier, P., Bernhofer, C., Buchmann, N., Gilmanov, T., Granier, A., Grunwald, T., Havrankova, K., Ilvesniemi, H., Janous, D., Knohl, A., Laurila, T., Lohila, A., Loustau, D., Matteucci, G., Meyers, T., Miglietta, F., Ourcival, J.-M., Pumpanen, J., Rambal, S., Rotenberg, E., Sanz, M., Tenhunen, J., Seufert, G., Vaccari, F., Vesala, T., Yakir, D., and Valentini, R.: On the separation of net ecosystem exchange into assimilation and ecosystem respiration: review and improved algorithm, Global Change Biol., 11, 1424–1439, https://doi.org/10.1111/j.1365-2486.2005.001002.x, 2005.